# Neural signatures of temporal anticipation in human cortex represent event probability density

Matthias Grabenhorst ®[1,2] ✉, David Poeppel ®[3] & Georgios Michalareas[1,2,4]

Temporal prediction is a fundamental function of neural systems. Recent results show that humans anticipate future events by calculating probability density functions, rather than hazard rates. However, direct neural evidence for this hypothesized mechanism is lacking. We recorded neural activity using magnetoencephalography as participants anticipated auditory and visual events distributed in time. We show that temporal anticipation, measured as reaction times, approximates the event probability density function, but not hazard rate. Temporal anticipation manifests as spatiotemporally patterned activity in three anatomically and functionally distinct parieto-temporal and sensorimotor cortical areas. Each of these areas revealed a marked neural signature of anticipation: Prior to sensory cues, activity in a specific frequency range of neural oscillations, spanning alpha and beta ranges, encodes the event probability density function. These neural signals predicted reaction times to imminent sensory cues. These results demonstrate that supra-modal representations of probability density across cortex underlie the anticipation of future events.

The anticipation of future events is foundational to many complex functions, such as associative learning[1–3], decision-making[4,5], and motor-preparation[6,7]. For example, a predator predicts its prey's movements, decides to quickly attack, and secures the next meal. A human boxer anticipates her opponent's actions, moves rapidly, and evades the attack. In communication, humans and animals track, predict, and quickly react to their partners' auditory and visual signals. Accordingly, temporal prediction is considered an elementary function of neural systems[8–10].

Temporal prediction requires the estimation of elapsed time relative to a reference point and the estimation of event probability over time. Understanding the brain mechanisms that solve this computational problem has been an important goal in neuroscience[11–18]. Still, the cortical representation of temporal event statistics is not understood.

Neural signals reflecting basic temporal prediction are observed across the cortical hierarchy. These signals are often associated with enhanced sensory perception and improved motor performance. In rodent primary auditory cortex[9] and non-human primate's early (V1)[19] and late (V4)[20] visual areas, as well as premotor cortex[5], neuronal firing rates reflect the expectation of events in time. Likewise, in humans, expectations modulate activity in auditory[21] and visual[22] cortices. Specifically, spectral power modulations in the alpha (8–13 Hz)[23] and beta (15–30 Hz)[17] bands are related to prediction in (sensori-)motor activity.

It is debated how such expectation-related signals emerge in sensory and motor regions[4,12]. There is evidence that prediction may, at least partially, be an intrinsic property of sensory systems[24,25] that, in addition to top-down modulatory effects[26], shapes perception. In motor cortex function, on the other hand, expectation-related signals critically depend on sensorimotor transformation[27].

The posterior parietal cortex (PPC) is essential for motor preparation, based on its connectivity to both sensory and action-

[1]Department of Cognitive Neuropsychology, Max-Planck-Institute for Empirical Aesthetics, Frankfurt, Germany. [2]Ernst Strüngmann Institute for Neuroscience in Cooperation with Max Planck Society, Frankfurt, Germany. [3]New York University, 6 Washington Place, New York, NY, USA. [4]CoBIC, Medical Faculty, Goethe University, Frankfurt, Germany. ✉e-mail: m.g@ae.mpg.de

planning systems[28,29]. An influential hypothesis of temporal prediction proposes a specific computation in PPC based on the hazard rate (HR) of events[11,30,31]. The HR represents the probability that a future event is imminent, given that it has not yet occurred[30]. The HR is widely assumed to be the canonical computation underlying temporal anticipation[3,11–17,19,20,32–35]. Specifically, the HR is argued to be represented in the lateral intraparietal area LIP in non-human primates, potentially driving saccades to anticipated future events[11]. In humans, HR computation is associated with several cortical areas, including the early visual cortex, inferior parietal lobule[32], and motor cortex[17,32]. However, the computation of HR is a complex[36] and numerically unstable[37] mathematical operation, which raises questions about its neurobiological plausibility and raises doubts about HR's generality as a neural prediction signal. This highlights a long-standing question: What is the cortical representation of event probability across time?

The probability density function of events over time, the event PDF, is a computationally simpler variable than the HR. The event PDF can be empirically approximated by two basic operations: the estimation of elapsed time and the counting of events over time, as more and more events are registered (Fig. 1a). Recently, a model of anticipatory behavior based on the event PDF was proposed[38,39].

The current study aimed to identify the cortical representation of temporal statistics by investigating HR-based and PDF-based models of anticipation to adjudicate between them. We are guided by two general hypotheses: (i) Inference of the temporal statistics of sensory events leads to a neural representation of probability over time. (ii) This representation of sensory statistics ultimately shapes a (fast) motor response, the behavioral hallmark of successful prediction[30,40]. Specifically, we assume an inverse relationship between event probability and motor preparation: when event probability is high, reaction time should be short, and vice versa (Fig. 1b).

We conceptualize temporal anticipation as a computational mapping problem where a neural representation of temporal statistics relates stochastic sensory input to anticipatory behavior, expressed as reaction time (RT) modulation (Fig. 1c). We performed experiments in audition and vision, two modalities that track the environment with high temporal resolution[41], to probe whether a modality-independent computation in higher-order cortex underlies temporal prediction. Building on our previous work[38,39], we relate the foundational computation of event probability over time, as captured by our recently proposed model of temporal anticipation, to neural data.

Here, we show that the probability density of sensory events is represented in specific posterior parietal and temporal areas and in the sensorimotor cortex before a sensory event occurs. These anticipation signals in the alpha and beta ranges predict reaction times to future auditory and visual cues.

## Results

We investigated neural activity during temporal anticipation with magnetoencephalography (MEG) while participants performed a Set-Go task in audition and vision (Fig. 2a). In the task, a Set cue was followed by a Go cue. The time between Set and Go, the Go time, was randomly drawn from truncated exponential or flipped exponential distributions (tau = 3.0303 for both), the Go time distributions, in separate blocks of trials (Fig. 2b). Participants were asked to press a button as fast as possible in response to the Go cue, generating RTs. In the task, fast reactions (short RT) represent strong event anticipation based on probability estimation, linking the experiment to many everyday tasks that require fast actions based on the prediction of future events.

23 participants performed the Set-Go task and generated 31,962 RTs (Fig. 2c). The RTs displayed typical features of a simple RT task[30]: the RT histogram showed a steep left flank and a short right tail, mean RT was short with small variance (0.261 s, SD = 0.085 s), 97.3% of RTs fell inside of the interval RT = [0.1, 0.5] s. Mean RT was shorter in the auditory than in the visual conditions ($-0.032 \pm 0.031$ s, mean ± SD, $P = 0.0076$, $t_{(22)} = -2.8$). RT variance was positively correlated with mean RT in all experimental conditions (Supplementary Fig. 1). Across the four conditions, there were an average of 1.59% of early (pre-Go cue) responses which were removed from the analysis. The number of early responses did not differ significantly between sensory modalities or between Go time distributions (Supplementary Tables S1, S2, and Supplementary Fig. 2).

The Go time distributions modulated auditory and visual RTs in a systematic way: when the Go cue probability was large, RT was short, and vice versa (Fig. 2d and e). As the Go cue probability decreased over time, RT increased (exponential conditions, Fig. 2d, left and 2e, left) and vice versa (flipped exponential conditions, Fig. 2d, right and 2e, right). The similarity of these RT dynamics across audition and vision is consistent with similar computations underlying the generation of RT in both modalities. To investigate these computations, we fit the RT curves with PDF-based models (Fig. 2d and e) and HR-based models (Fig. 2f and g), which are described in detail in the next section.

### Models of anticipation based on probability estimation

For a long time, it was postulated that the critical variable underlying temporal anticipation, i.e., relating stochastic sensory input to RT output, is the HR of events[11–17,19,20,32,33]. Recent work challenged this hypothesis by demonstrating that humans estimate the event PDF but not the HR[38,39]. Here, both hypotheses were examined in order to verify that the better fit of the PDF-based model over the HR-based model to the behavioral data is replicated.

The functional form of the model of anticipatory behavior has specific implications for underlying neural dynamics. The HR-based model accounts for two sources of uncertainty with specific hypotheses about their functional form. The first source is the uncertainty in time estimation, which is hypothesized to linearly increase with elapsed time ("scalar property")[42,43]. This hypothesis follows a simple intuition: the accuracy of time estimation linearly decreases with the length of the to-be-estimated time interval. The second source of uncertainty is the distribution of the event occurrence probability across time, the event PDF.

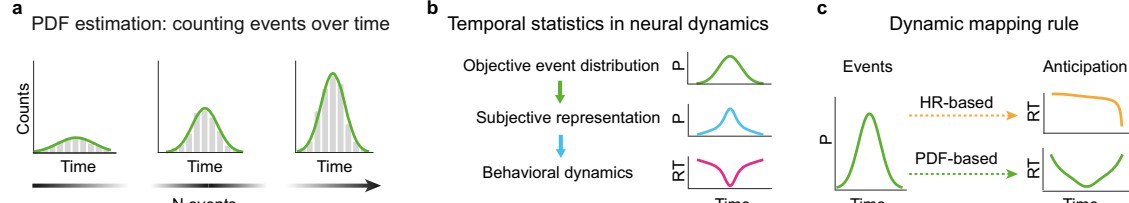

**Fig. 1 | Concepts and hypotheses in temporal anticipation. a** Neural systems may estimate the event probability density function (PDF) by counting events over time. **b** A subjective representation of temporal event statistics, e.g., probability over time, leads to adaptive behavior: where event probability is high, reaction time is short, and vice versa. **c** Temporal anticipation as a computational mapping problem. To adjudicate between computational hypotheses, the objective event PDF is related to reaction time, a proxy for anticipation, by different mapping rules (hazard-rate-based and PDF-based, Methods).

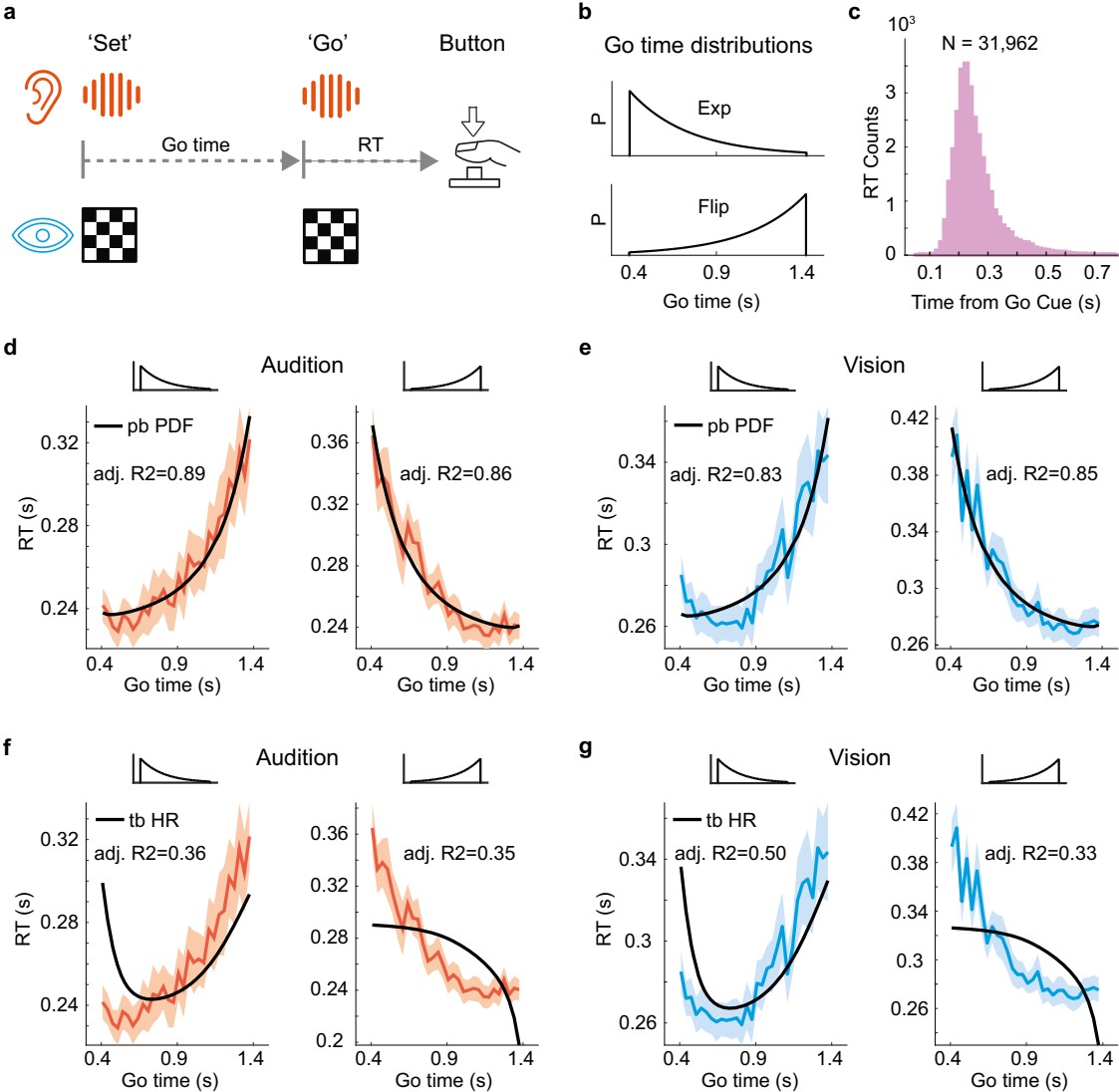

**Fig. 2 | Participants estimate the probability density function of future events: temporal anticipation. a** Set-Go task schematic. In auditory and visual blocks of trials, participants were asked to respond as fast as possible to the Go cue, generating a reaction time (RT). In auditory blocks of trials, white noise bursts of 50 ms duration served as Set and Go cues; in visual blocks of trials, checker boards of 50 ms duration served as cues (Methods). **b** In separate blocks of trials, the time between Set and Go (Go time) was drawn from a truncated exponential distribution (Exp) or from its left-right flipped version (Flip). **c** Histogram of RT pooled across the 23 participants and all experimental conditions. **d** Reciprocal probabilistically blurred PDF fits mean auditory RT over Go time (Methods). **e** Reciprocal probabilistically blurred PDF fits mean visual RT. **f** Mirrored temporally blurred HR (Methods) does not capture auditory RT. **g** Mirrored temporally blurred HR does not capture visual RT. **d**–**g** For plotting, RTs and fit curves were smoothed by reducing the Go time step size from 16 to 32 ms. Shaded areas represent s.e.m. across participants. Source data are provided as a Source Data file.

The brain's estimate of the event PDF is hypothesized to be affected by the uncertainty in elapsed time estimation. In modeling, this is instantiated by a convolution of the objective event PDF with a Gaussian blurring kernel representing the uncertainty in elapsed time estimation, a computation we refer to as temporal blurring (Fig. 3a, left).

The HR-hypothesis implies that neural systems transform the temporally blurred event PDF to a more complex form, the Hazard Rate (HR). Mathematically (and computationally), this requires several sequential steps: First the estimation of the PDF, $f(t)$, then the estimation of its integral across time for computing the cumulative density function (CDF), $F(t) = \int_{-\infty}^{t} f(u)du$, and, lastly, subtraction and division for estimating the HR as $h(t) = \frac{f(t)}{1-F(t)}$ (Fig. 3a, middle).

Finally, the HR model hypothesizes an inverse relationship between the neural representation of HR and a motor response; i.e., when HR assumes large values, anticipation is high, predicting short RT

and vice versa. This relationship is expressed by a linear mirroring of the HR variable around a fixed value (Fig. 3a, right).

The PDF-based model contrasts in three ways from the HR model. First, it postulates that the human brain represents the event probability across time in a computationally simpler form than the HR, i.e., by estimating the PDF itself. Second, in the PDF-based model, the uncertainty in elapsed time estimation does not increase monotonically with time, but it is modulated by the event probability density: when event probability is large, temporal estimates are precise and vice versa. This concept is termed probabilistic blurring[38,39] (Fig. 3b, left, Methods). Third, the inverse relationship between probability and RT is assumed to be reciprocal[38,39] (Fig. 3b, right).

## Event PDF computation underlies anticipatory behavior

Both the HR-based variable (temporally blurred mirrored HR, Methods) and the PDF-based variable (probabilistically blurred reciprocal

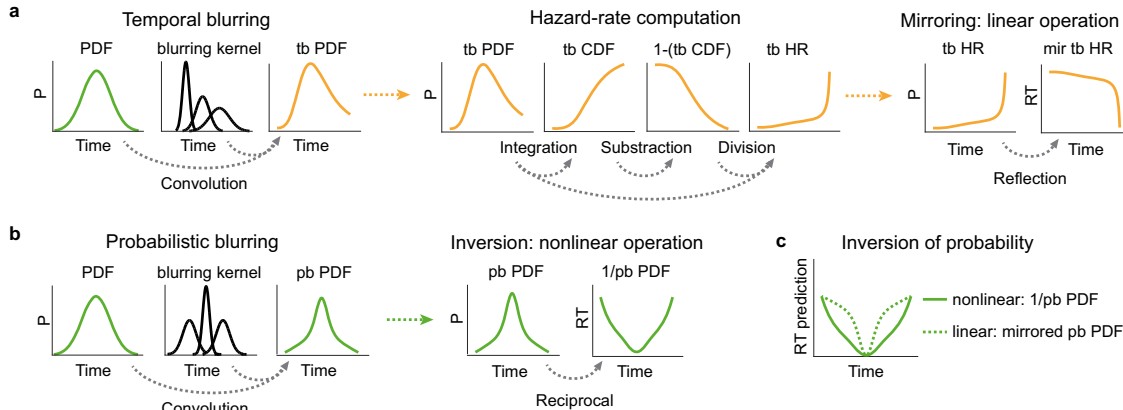

**Fig. 3 | Computational hypotheses in temporal anticipation. a** Canonical hazard rate hypothesis. Left: temporal blurring ("tb") poses that uncertainty in the neural estimation of elapsed time increases linearly with elapsed time, affecting the estimate of event PDF (Methods). Middle: HR computation requires a sequence of three mathematical operations (Methods). Right: The HR hypothesis predicts a linear, mirrored ("mir") relationship between event probability over time and anticipation (Methods). **b** PDF-based hypothesis. Left: Probabilistic blurring. Event PDF determines uncertainty in elapsed time estimation. In modeling, the objective event PDF is convolved with a Gaussian kernel whose SD is inversely related to the event PDF, resulting in a subjective estimate of the event PDF: the probabilistically blurred ("pb") PDF (Methods). Right: Reciprocal probabilistically blurred PDF ("1/pb PDF") predicts RT. **c** Effect of reciprocal computation on RT prediction. 1/pb PDF implies a nonlinear "weighting" of event probability in temporal anticipation. For comparison, probabilistically blurred PDF was linearly inverted ("mirrored pb PDF") to illustrate the effect of the presumed nonlinear reciprocal computation. For comparison, both 1/pb PDF and mirrored pb PDF were scaled by their respective ranges. The *y*-axis in arbitrary units.

PDF, Methods) were computed and fit to the RT data (Supplementary Data 1). The PDF-based model captured the RT dynamics well in audition (Fig. 2d) and vision (Fig. 2e), and in both exponential and flipped exponential conditions, as supported by large values of adjusted $R^2$. In contrast, the fits of the canonical HR-based model failed to capture the RT dynamics (Fig. 2f and g). Each HR-based fit resulted in a substantially lower value of adjusted $R^2$ compared to the PDF-based model. In the flipped exponential case, the HR-based model even made the qualitatively wrong prediction of a convex RT shape when the RT curve was in fact, concave (Fig. 2f and g, right). For completeness, we evaluated the effect of the two different blurring concepts on the model fits. We computed two more variables (probabilistically blurred mirrored HR and temporally blurred reciprocal PDF) and fit them to RT data (Supplementary Fig. 3). This additional analysis shows that irrespective of the type of blurring, the PDF-based models outperform the HR-based models, replicating our previous results[39].

There was no evidence from behavior that humans compute the HR to predict the timing of future events. Instead, the analysis of the psychophysical data indicates that participants estimate a computationally simple variable, the event PDF, to predict the timing of future sensory cues.

## Alpha and beta power decrease before an anticipated event

We sought to identify the neural signatures of anticipation in cortical population activity using MEG. We focused the analysis on the time span preceding the Go cue (Fig. 2a) since this is when a representation of the event PDF should be reflected in neural activity. This is based on the assumption that neural activity before an anticipated cue may potentiate the neural circuitry that will be activated by the cue, leading to an earlier onset of motor response and shorter RTs.

The MEG data were aligned to the Set and to the Go cues. In each alignment, the time-frequency decomposition was computed at each time point of each trial using a moving window (Hanning). We first investigated power changes prior to the Go cue relative to a pre-Set cue baseline (−0.65 to −0.25s).

Before the Go cue, power decreased significantly in the alpha (7–12 Hz) and beta (15–35 Hz) frequency bands (Fig. 4a, $P = 0.001$, cluster test, Methods). Such differences in neural activity between pre-

Set and pre-Go periods suggest activity related to Go cue anticipation, based on task demands.

The relative decrease in alpha power was located in two sensor areas (Fig. 4b, top left). One location comprised sensors over left motor areas, in line with sensorimotor alpha (mu)[44]. This raises the question whether sensorimotor alpha plays a role in temporal prediction, which we address later with a correlation analysis. The other location comprised right posterior sensors. In the case of spatial attention, lateralization of posterior alpha power is commonly reported, i.e., power increases ipsilateral to the direction of attention (expected stimulus direction) and decreases contralaterally. This is often observed in visual[45] and, in a similar fashion, in auditory tasks[46]. The present study, however, does not feature spatial lateralization as a component: stimulation in vision and audition was bilateral. Therefore, the rightward lateralization observed here does not indicate a spatial attention phenomenon.

Alpha power was consistently suppressed prior to the event, with the strongest desynchronization around 150 ms before the Go cue. Alpha suppression was progressively diminished after that, reaching zero at the time of the Go cue (Fig. 4b, bottom left). These dynamics suggest an anticipation signal.

The relative decrease in beta power was located in sensors over the left motor areas (Fig. 4b, top right). Beta power was also consistently suppressed before the Go cue, with the maximum suppression occurring during the last 200 ms before the Go cue and then returning towards zero. Again, this suggests an association to prediction (Fig. 4b, bottom right).

Finally, we observed an increase of low-frequency power (3–7 Hz, Fig. 4a) just after the Go cue occurrence. This reflects the signatures of the event-related field transient activity[47], which also extended to higher frequencies.

## Sensor-level correlation analysis scheme

We investigated how the observed alpha and beta band neural dynamics relate to the temporal anticipation of the Go cue. We computed Spearman's correlation between power in each MEG time-frequency-sensor triplet (Fig. 5a, top) and RT at the single-trial level (Fig. 5a, bottom, Methods). A cluster-based permutation test on Spearman's rho identifies clusters in time-frequency-sensor space in which rho significantly differed from zero. Importantly, this cluster test was set up to run across

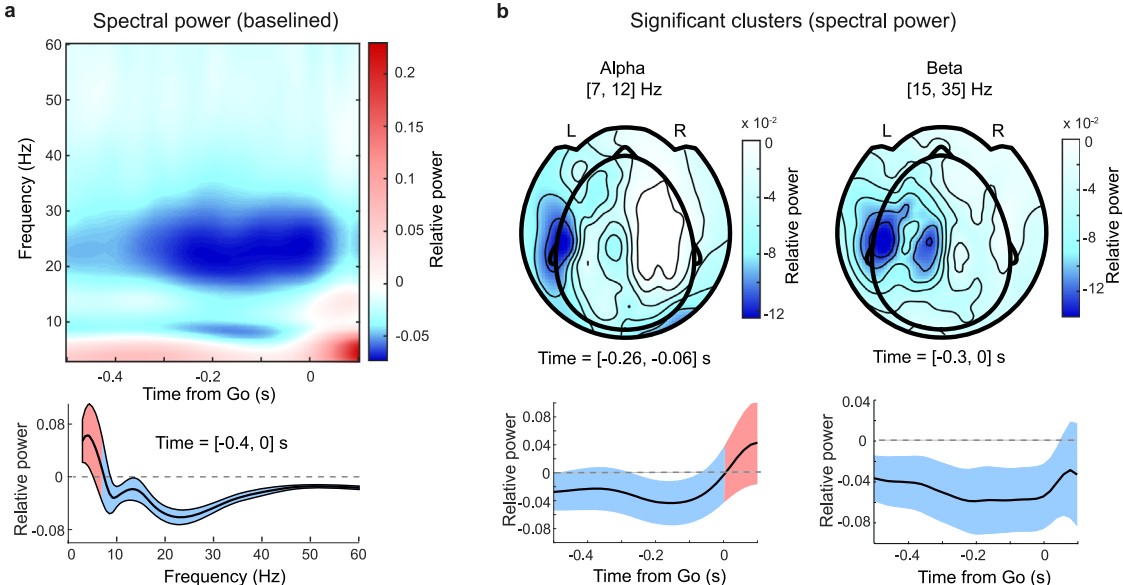

**Fig. 4 | Alpha and beta band power decrease before anticipated Go cue. a** Top: Spectral power averaged across participants, conditions (audition and vision, exponential and flipped exponential distributions), and MEG channels, baselined relative to a pre-Set cue time span ($t = [-0.65, -0.25]$ s, Methods). Desynchronization in alpha (7–12 Hz) and beta (15–30 Hz) bands prior to Go cue. Bottom: Decrease in alpha and beta spectral power changes averaged across channels, and time ($t = [-0.4, 0]$ s), shaded area depicts SD across time. **b** Top: Sensor array topography of significant desynchronization in alpha (left) and beta (right) power (cluster test, power against zero). L and R labels indicate the left and right hemispheres. Bottom: mean power change averaged across channels, shaded areas depict SD across channels. Source data are provided as a Source Data file.

the two sensory modalities (audition and vision) and across the two Go time distribution conditions (exponential and flipped-exponential). Thus the test was specified to separately search for significant positive and negative correlation clusters in time-frequency-sensor space that are shared by all four experimental conditions.

This analysis strategy can identify neural activity prior to the Go cue that is correlated with RT. Note that RT demarcates the time point of the button press after the Go cue. Therefore, a significant correlation indicates that neural activity before the Go cue is predictive of RT after the cue. Since RT is driven by event probability, as demonstrated by the convincing fits of the PDF-based RT model (Fig. 2d and e), the correlation between pre-Go-cue spectral power and RT is indicative of a neural representation of the event PDF.

**Sensor-level alpha and beta power represent event PDF**
The correlation analysis between spectral power and RT identified a negative correlation cluster that comprised both alpha and low beta bands, namely the range between 7 to 22 Hz (Fig. 5b, top, $P = 0.001$). Notably, the dynamics of Spearman's rho over frequency (Fig. 5b, bottom) displayed two distinct negative peaks (troughs) in the alpha and low beta ranges (alpha: $f = 7.9$ Hz, low beta: $f = 17.0$ Hz, Supplementary Analysis). This supports a within-cluster separation into these two bands. The cluster was split according to alpha (7 to 12 Hz) and beta (15 to 22 Hz) frequency ranges, each comprising a set of MEG sensors that were selected for plotting.

Topographically, the negative alpha band correlation was right lateralized over right posterior sensors (Fig. 5c, top). No corresponding negative correlation was found over the left sensorimotor areas. Within the negative alpha cluster's channels, in the time span before the Go cue, Spearman's rho decreased over time from $-0.3$ s to 0 s and reached its largest negative value immediately before the Go cue (Fig. 5c, bottom). This indicates that alpha power was most predictive of RT at the time point of Go cue occurrence, suggesting a functional role of alpha oscillations in temporal anticipation. Importantly, the correlation between alpha power and RT was not driven by eye blink

activity as demonstrated in control analyses (Supplementary Figs. 4–7, Methods).

The negative correlation in the low beta band was also located over the right posterior sensors and extended towards the right frontal sensors (Fig. 5d, top). Between $-0.4$ and 0 s relative to the Go cue, Spearman's rho remained consistently negative. Immediately after the Go cue, rho increased towards zero (Fig. 5d, bottom), similar to the correlation in the alpha band. This dynamic entails that low beta power is most predictive of RT right until Go-cue occurrence, hinting towards a functional role of low beta oscillations in temporal anticipation.

Additionally, a positive correlation cluster was observed in the high beta (23–35 Hz) band (Fig. 5b). This cluster was left lateralized and was located over left sensorimotor areas (Fig. 5e, top). Spearman's rho averaged across the cluster's sensors, displayed an interpretable dynamic: Rho increased from $-0.4$ to $-0.1$ s relative to the Go cue, peaked at around $-0.1$ s, and thereafter rapidly decreased, reaching a minimum at the time-point of Go cue occurrence (Fig. 5e, bottom). At -0.2 s after the Go cue, rho reached its peak and rapidly decreased towards zero thereafter. In light of the distribution of RTs (Fig. 2c), which has a grand mean of RT = 0.261 s, the correlation coefficient peaks ~60 ms before the button press, suggesting that the modulation of high beta power is related to motor execution. This interpretation is in line with the common finding of a beta band desynchronization during the preparation and execution of movements[48].

Taken together, the sensor-level analyses identified three frequency bands of interest in which spectral power prior to the Go cue was correlated with RT. Importantly, two of the neural signals seem linked to prediction: alpha band correlation decreased, and high beta band correlation increased until just before the Go cue, suggesting a functional relationship between spectral power and temporal anticipation. Low beta band correlation also displayed an interpretable dynamic: its power was most predictive of RT before the Go cue occurrence. The fact that spectral power in the three frequency bands before the Go cue correlates with RT, i.e., the time point of the

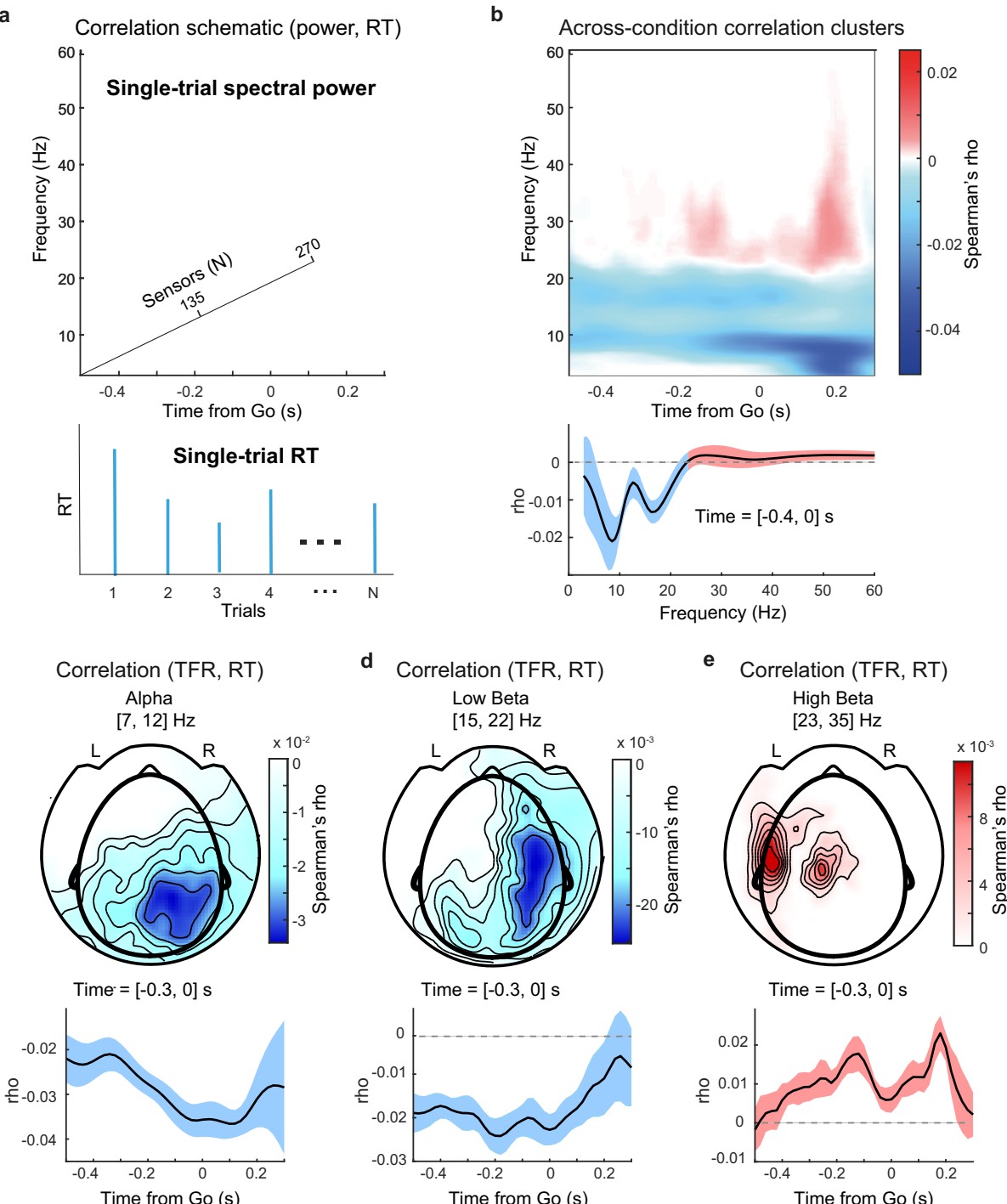

**Fig. 5 | Sensor-level alpha and beta power predict reaction time before antici-pated events. a** Correlation analysis schematic: within each experimental condi-tion, single-trial Spearman correlation is computed between spectral power in each time-frequency-sensor triplet (not baselined) and RT. **b** Top: Negative correlation cluster comprises alpha and low beta frequency bands ($P = 0.001$), positive corre-lation cluster in the high beta band ($P = 0.087$) (across-condition cluster test, Methods). Bottom: Mean Spearman's rho across time and sensors, shaded area depicts SD across time. **c** Top: Sensor-level topography of alpha-band correlation cluster. Bottom: Mean rho and SD across cluster sensors. Before the Go cue, rho decreases until right before the Go cue. **d** Sensor-level topography of low beta band correlation cluster. Bottom: Mean rho and SD across cluster sensors. Rho increases right after the Go cue. **e** Sensor-level topography of high beta band correlation cluster. Bottom: Mean rho and SD across cluster sensors. Rho increases until right before the Go cue, then decreases, reaching a local minimum at the time of the Go cue. L and R labels indicate the left and right hemispheres. Source data are provided as a Source Data file.

button press *after* the event, indicates that a neural representation of the event PDF underlies the anticipation of sensory cues. In order to better understand the functional role of these neural dynamics, we next aimed to reveal the cortical localization of the three correlation signals.

## Source-level correlation analysis scheme
Within each frequency band-of-interest (7–12 Hz, 15–22 Hz, 23–30 Hz), identified in the above sensor-level analysis, spectral power was projected to brain source-space using a spatial filter (DICS beamforming[49], Methods). This source space for each participant

comprised the individual's cortical mantle representation, as extracted from a structural MR scan. This source space was fused with a whole-cortex parcellation atlas (AAL atlas[50], Methods) in order to relate findings to known anatomical areas and assist the interpretation of results. Within each brain source, power in each frequency band of interest was averaged across its frequencies. Spearman correlation was then computed between single-trial average power and single-trial RT in each source separately. Finally, a cluster test across all experimental conditions (audition and vision, exponential and flipped exponential Go time distributions) identified the cortical sources corresponding to the significant correlation in the frequency bands of interest (Supplementary Methods). This across-condition approach identifies correlation signals common to all experimental conditions. To investigate correlation dynamics across time, this analysis was run within each of eight-time windows relative to the Go cue, centered at $t = -0.35$ to $t = 0.35$ seconds in steps of 100 ms (Methods).

We note that this analysis in source space was used to localize the effects already identified at the sensor level, where the null hypothesis had already been rejected. The cluster-based permutation test on source-space data was not used to reject again the null-hypothesis. The test was used to identify the cortical areas where the correlation was significantly different from zero in each time window and the frequency band of interest identified at the sensor level. When projecting the data from the sensor space into source space, all the sensors are used, not only the subset that belongs to the sensor-level cluster.

### Alpha band power in right IPL and pMTG predicts RT
The source-level correlation between alpha (7–12 Hz) power and RT was right lateralized (Supplementary Fig. 8), covering posterior parietal and posterior temporal cortices (Fig. 6a top row). Spearman's rho decreased prior to the Go cue and reached the largest negative values around 50 ms before the cue (Fig. 6a top row, Fig. 6b), confirming the dynamics observed in sensor-level analysis (Fig. 5c, bottom).

To demonstrate more precisely the anatomical distribution of the correlation signal over time, we re-plotted the same analysis using an individual colormap for each time window (Fig. 6a bottom row and Supplementary Fig. 9). This revealed that at −0.35 s before the Go cue, the correlation signal has prominent local maxima in right supramarginal gyrus (SMG) and posterior middle temporal gyrus (pMTG). Beyond this early time window, the signal quickly spread out across the entire right inferior parietal lobule (IPL) at ~−0.25 s and extended to the posterior MTG region at −0.15 s. After the Go cue, the correlation decreased in cortical extent (Fig. 6c). This source-level analysis thus identified two cortical regions, one including the right IPL, the other the right pMTG, in which the dynamics of alpha-band oscillations followed an approximation of the event PDF before the occurrence of an anticipated event. These findings suggest a supra-modal, functional role of the right IPL and right pMTG area in temporal event anticipation.

### Low beta band power in right SPL predicts RT
The source-level correlation between low beta power (15–22 Hz) and RT was right lateralized in six time windows spanning −0.35 to 0.15 s relative to the Go cue (Fig. 6d). In this time range, the negative correlation signal covered parts of right PPC but not occipital cortex. Notably, the correlation signal was strongest in the right superior parietal lobule (SPL) at −0.05 s, i.e., just before the Go cue occurrence (Fig. 6e and f). The analysis identified the right SPL as the cortical site where low beta power is most predictive of RT before the anticipated cue. This relationship between low beta oscillations and RT indicates a representation of the event PDF in cortical dynamics, suggesting a supra-modal, functional role of right SPL in temporal anticipation.

### High beta band power in left sensorimotor cortex predicts RT
The positive correlation in the high beta band (23–30 Hz) localized to left sensorimotor cortex (Fig. 7a), in agreement with the right index finger button press. The correlation increased in magnitude over time, peaked at 0.05 s post cue, and decreased thereafter (Fig. 7b). The cortical extent of the correlation signal, expressed as the number of brain sources, displayed a similar dynamic (Fig. 7c), it increased prior to the Go cue and decreased quickly thereafter.

The positive sign of the correlation signal is in line with a hypothesized disinhibitory role of beta band desynchronization[48]: small power values predicted short RTs and vice versa. This interpretation is supported by the fact that the correlation signal increased before participants pressed the response button. Interestingly, the positive correlation between RT and beta power occurred hundreds of milliseconds before the anticipated sensory event (Go cue), demonstrating that motor preparation itself follows an approximation of the event PDF.

The correlation analysis between spectral power and RT reveals that, before anticipated auditory and visual sensory events, alpha, low beta, and high beta power dynamics represent an approximation of the event PDF. This conclusion is based on two arguments. First, the RTs, which are used as a regressor on spectral power, are fit by the reciprocal event PDF (Fig. 2d and e). This implies a specific, nonlinear relationship between RT and the event PDF. The observed correlation between RT and spectral power thus indicates that the reciprocal event PDF is approximated by neural dynamics. Second, the RTs occur after the Go cue, yet the analysis identified correlations between RT and spectral power before the Go cue. This indicates that neural activity represents an approximation of the event PDF before the Go cue – the hallmark of a neural prediction signal.

Taken together, these three main results link the specific computation of event PDF to right-lateralized cortical dynamics and demonstrate a supra-modal role of cortical areas SPL, IPL, pMTG, as well as left sensorimotor cortex in the prediction of future events.

### Time-frequency representations of event probability density
In an additional time-frequency decomposition analysis, we aimed to identify a direct representation of the event PDF in neural dynamics. Based on the convincing fits to group-level RT (Fig. 2d and e), we fit the PDF-based variable (probabilistically blurred reciprocal PDF) to single-trial RT (Methods). This PDF-based model of RT was then Spearman-correlated with single-trial source-level power, averaged within each of the three frequency bands of interest (7–12 Hz, 15–22 Hz, 23–30 Hz). A cluster-based permutation test identified the sources in which Spearman's rho significantly differed from zero (Methods). This correlation analysis was performed within four-time windows relative to the Go cue (−0.4 to 0 s, window size 0.1 s).

This analysis replicated the cortical topography of the correlation analysis between spectral power and RT reported above. In the alpha (7–12 Hz) band, the analysis identified a correlation signal in the right IPL and the pMTG area (Supplementary Fig. 10a). In the low beta (15–22 Hz) band, the correlation signal was principally in the right SPL region (Supplementary Fig. 10b). In the high beta (23–30 Hz) band, the correlation signal covered left sensorimotor areas (Supplementary Fig. 10c). This analysis further demonstrates that neural dynamics in alpha and beta frequency bands can be interpreted to represent the event probability distribution over time prior to anticipated sensory events in specific temporoparietal areas and in sensorimotor cortex.

### Sensor-level ERF analysis scheme
We next investigated the potential effects of event probability density on neural activity time-locked to the Go cue, i.e., on the event-related fields (ERFs). The Go stimulus resulted in a prominent P1 response in both auditory (Fig. 8a top) and visual (Fig. 8a bottom) conditions, originating in primary sensory areas (topography plots, Fig. 8a). At the level of averaged data and based on visual inspection, the time course

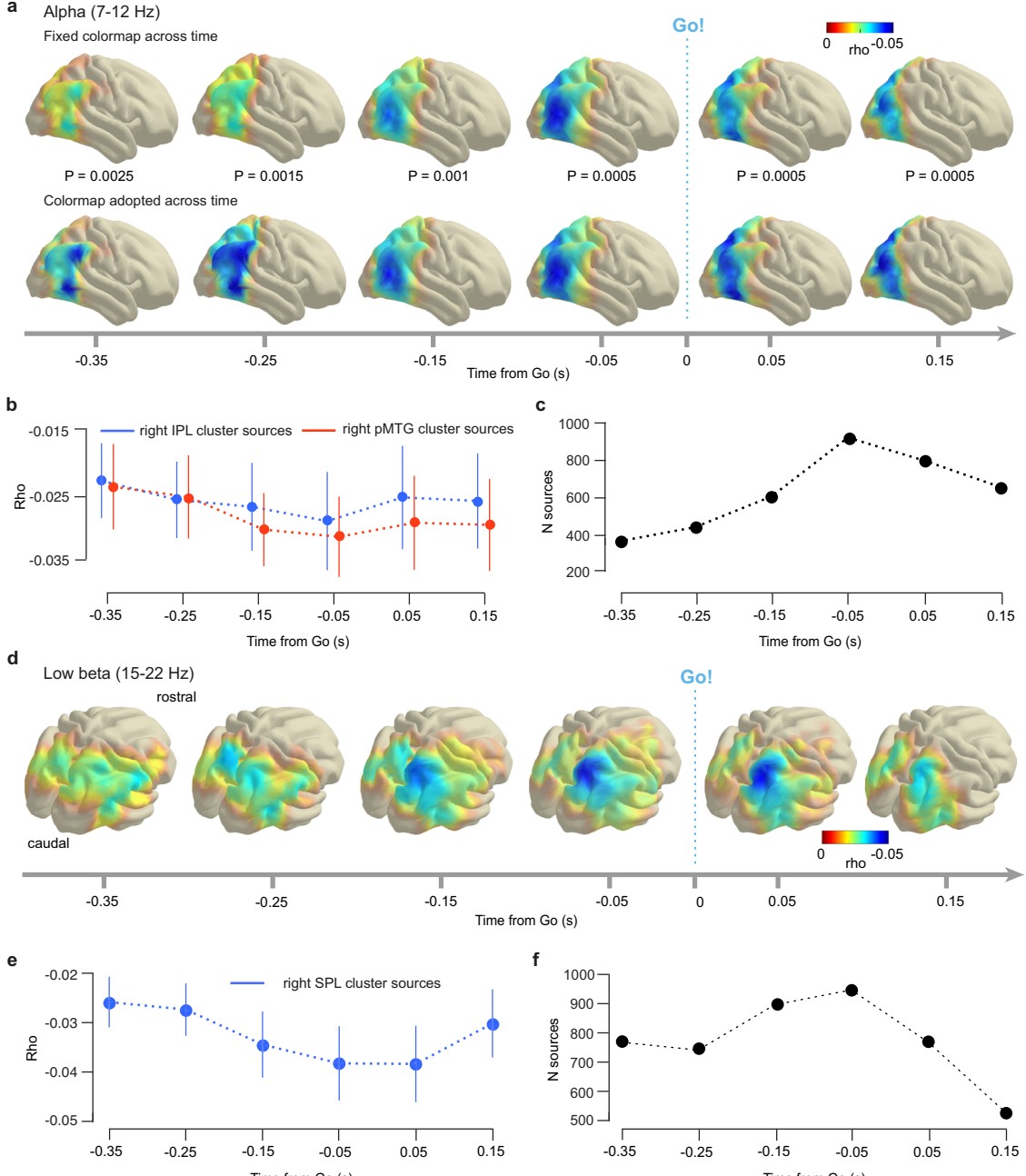

**Fig. 6 | Alpha band power in right parieto-temporal areas and low beta band power in right superior parietal lobule predict reaction time to anticipated events. a** Top row: Cortical topography of across-condition Spearman correlation (mean alpha (7–12 Hz) power, RT). Negative correlation reaches a maximum in the posterior middle temporal gyrus (pMTG) region immediately before the Go cue. Colormap fixed across time to highlight across-time correlation dynamics. Bottom row: Re-plotting of the top row with an individual colormap per time span (Supplementary Fig. 9) highlights the correlation extent for each time span. Negative correlation at − 0.35 s is in the supramarginal gyrus, and pMTG spreads out to cover the entire inferior parietal lobule (IPL) at −0.25 s and extends to the pMTG region from − 0.15 s on. For visualization, the average cortical surface of the 1200 Subjects Group Average Data[87] from the Human Connectome Project was used. **b** Mean Spearman's rho averaged across sources from two regions of interest (blue: right

IPL, $N = 85$, red: right pMTG, $N = 71$). Averaging is first done within participants and across sources, then across participants. Error bars are standard errors of the mean. **c** Number of correlation sources over time. **d** Cortical topography of across-condition Spearman correlation (mean low beta (15–22 Hz) power, RT). Negative correlation is maximal in the right superior parietal lobule (SPL) before the Go cue. All $P$-values = 0.0005, uncorrected, Methods. All colormaps fixed across time to highlight across-time correlation dynamics. For visualization, the average cortical surface of the 1200 Subjects Group Average Data[87] from the Human Connectome Project was used. **e** Mean Spearman's rho averaged across sources ($N = 28$) from right SPL. Averaging first done within participants and across sources, then across participants. Error bars are standard errors of the mean. **f** Number of correlation sources over time. $P$-values uncorrected. Source data are provided as a Source Data file.

of early neural activity (< 0.15 s) was similar across the exponential and flipped-exponential probability conditions in both vision and audition. We next aimed to quantify potential neural signatures of event probability density in time-locked data before the Go cue and in early and late ERF components after the Go cue.

We used a correlation-based approach to relate neural activity, time-locked to the Go cue, to RT. At the single-participant level, the MEG data and the RTs were aggregated within adjacent pairs of Go times (called frames from this point on) for mild smoothing. For each sensor-by-time-point duplet, Spearman's rho was computed between

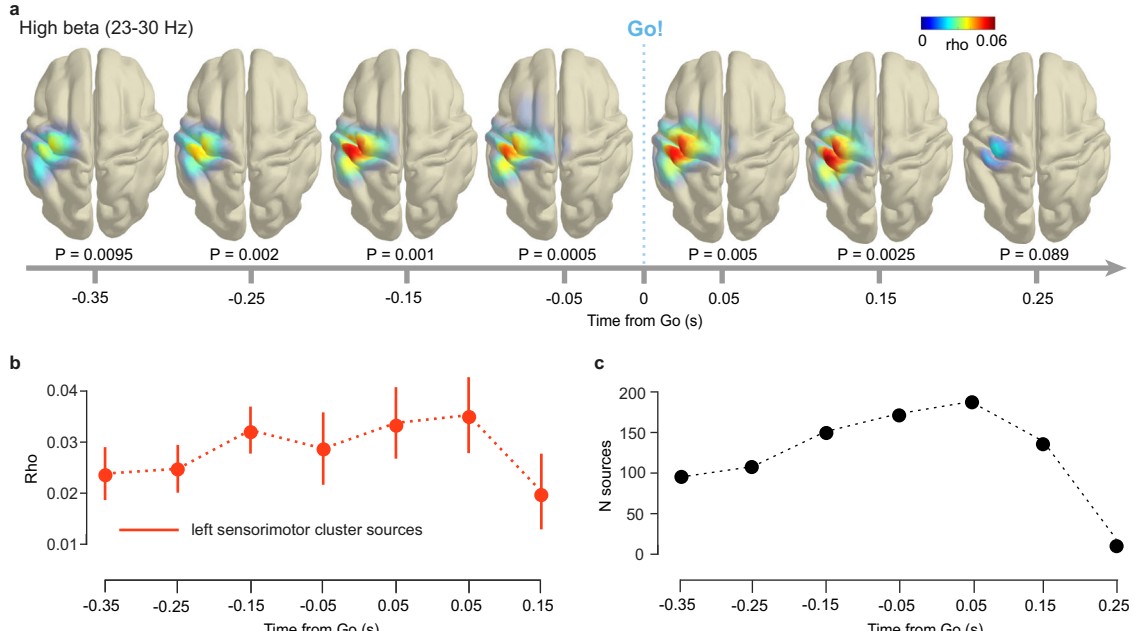

**Fig. 7 | High beta band power in left sensorimotor areas predicts reaction time to anticipated events. a** Cortical topography of across-condition Spearman correlation (mean high beta (23–30 Hz) power, RT. Positive correlation in left sensorimotor cortex prior to Go cue. All colormaps fixed across time to highlight across-time cluster dynamics. For visualization, the average cortical surface of the 1200 Subjects Group Average Data[87] from the Human Connectome Project was used. **b** Spearman's rho averaged across sources ($N = 119$) from the left sensorimotor cortex. Averaging first done within participants and across sources, then across participants. Error bars depict the standard error of the mean. **c** Number of correlation sources over time. *P*-values uncorrected. Source data are provided as a Source Data file.

the 30 frames of MEG data and the 30 averaged RTs, resulting in a correlation matrix (sensors by time points). On all participants' rho values, a cluster-based permutation test was run within each experimental condition to identify sensor-by-time clusters in which rho differs from zero (Methods). This analysis pipeline was run using a pre-Set baseline – 0.5 to 0 s) to investigate the time span before the Go cue and using a pre-Go baseline (– 0.5 to 0 s), to investigate the time span after the Go-cue.

### Sensor-level event-related fields correlate with RT
The analysis did not identify a significant correlation cluster in the time span before the Go cue. No correlation was observed in early sensory ERF components (< 75 ms) in visual or auditory areas. This indicates that either these sensory responses were not modulated by probability density or that our design was not sensitive enough to isolate these small effects, which may require a dedicated design that allows for averaging across several hundreds of trials per frame[51].

After the Go cue, two positive and two negative correlation clusters were identified. An early positive cluster (~ 0.25 s post-Go) and a late positive cluster (~ 0.4 s post-Go) occurred in all four experimental conditions (Fig. 8b, left). An early negative correlation cluster was identified (~ 0.25 s post-Go) in all four conditions and a late negative cluster (~ 400 ms post-Go) in the auditory exponential and the visual flipped exponential conditions (Fig. 8b, right). In order to better interpret the correlation, we selected two time points of interest (0.25 s and 0.4 s), that were shared by the positive and negative correlation clusters, for a source-level analysis.

### Source-level ERF analysis scheme
In the next analysis, we aimed to obtain a more precise spatial estimate of the correlations between ERFs and RT to better understand the functional role of time-locked neural activity in temporal anticipation. We employed a source-level correlation-based analysis. The per-frame MEG data (see above) were projected to source space

(LCMV beamforming, Methods), and Spearman's rho was computed between ERF and RT for each source and time point. Rho was averaged across time within windows-of-interest that were chosen based on the sensor-space clusters (30 ms windows centered around 0.25 and 0.4 s post Go) and one additional 30 ms window centered at 0.1 s post Go cue to probe for early correlations before the button response. We carried out across-condition cluster-based permutation tests to reveal source-level correlation clusters shared by audition and vision.

### Event-related fields in temporal anticipation
We observed a significant positive cluster over the right parietal cortex at $t = 0.25$ s post-Go (Fig. 8c, top left, $P = 0.012$). This cluster's largest rho values covered a right parietal area that included the right postcentral gyrus, anterior-lateral SPL, and intraparietal sulcus. The cluster's temporal and anatomical locations are in agreement with the ERP component P300, more specifically, with the P3b, which originates from parietal areas[51,52]. The positive correlation with RT implies that the P3b's amplitude negatively covaries with probability density: P3b amplitude is small where probability is high and RT is short, and vice versa. This is in agreement with the P3b's sensitivity to probability[53,54], more specifically, with work that promotes a negative correlation between P300 amplitude and probability[55]. Our results extend these findings to probability over time.

At 0.25 s post Go, we identified a second positive correlation cluster consisting of sources in the cerebellum (Fig. 8c, top right, $P = 0.005$). The positive correlation expresses that when cerebellar activity is small, RT is short, and vice versa. This finding is in agreement with the known functional role of the cerebellum in (finger[56]) movement control[57]. Alternatively, it may reflect activity related to the representation of time, an activity that is also linked to the cerebellum[58]. Both interpretations seem plausible, but a targeted experiment is needed to assess more conclusively the function of the cerebellum in temporal anticipation.

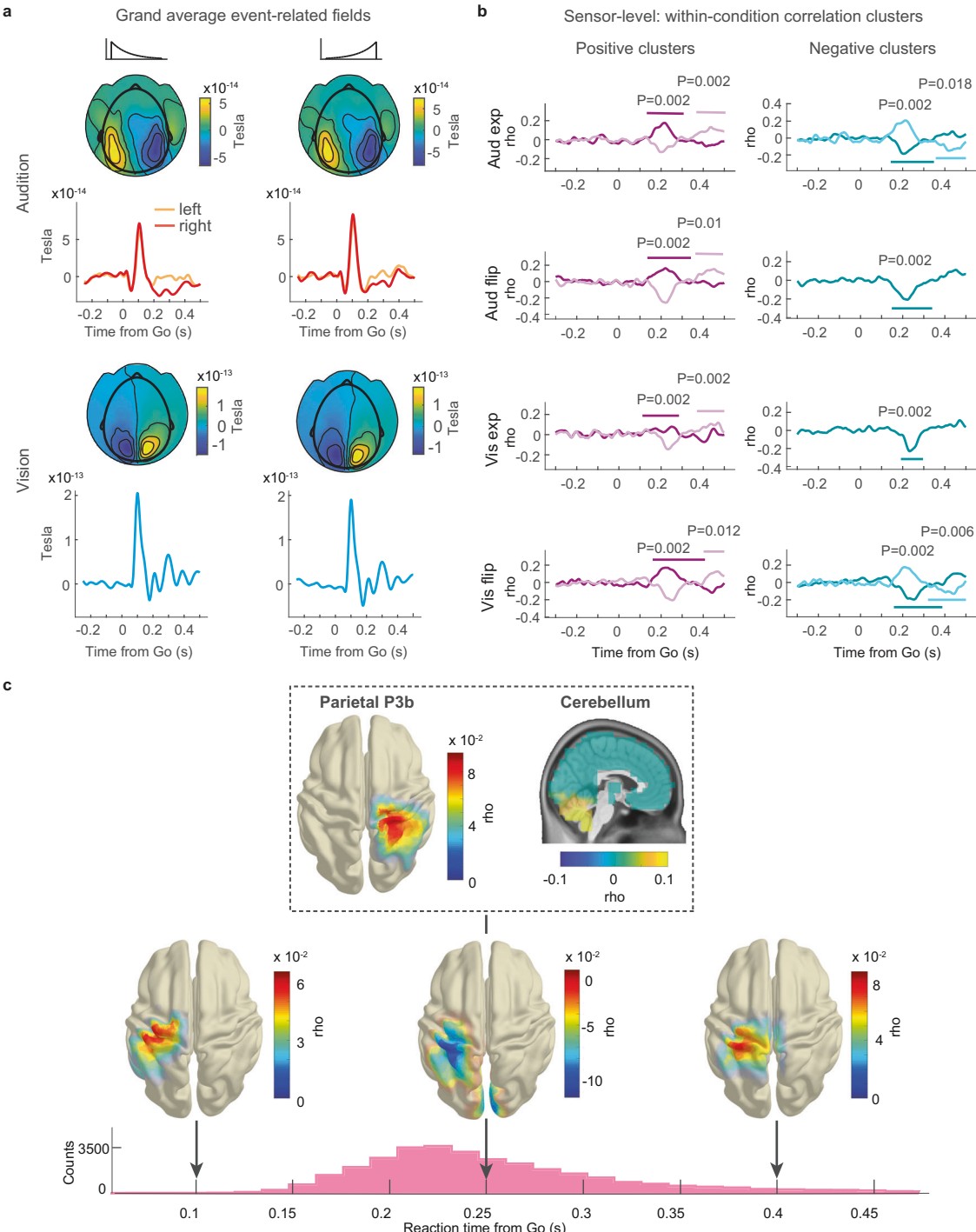

**Fig. 8 | Parietal, cerebellar, and sensorimotor event-related fields correlate with reaction times to anticipated events. a** Grand-average event-related fields plotted through participant-level selection of ERF sensors over auditory and visual areas. **b** Early and late positive and negative sensor-level correlation clusters (ERF, RT). Mean Spearman's rho averaged across clusters' sensors over time. Colored lines demarcate cluster time spans, P-values uncorrected. **c** Topography of across-condition source-level Spearman correlation clusters (ERF, RT). At 0.25 s post-Go cue, positive correlation clusters in the right parietal cortex (P3b) and cerebellum and a negative cluster in the left sensorimotor cortex. At 0.1 s and 0.4 s, positive correlation clusters in left sensorimotor cortex. Bottom: RT histogram from Fig. 2c for orientation. Source data are provided as a Source Data file. For visualization, the average cortical surface of the 1200 Subjects Group Average Data[87] from the Human Connectome Project was used.

At 0.1, 0.25, and 0.4 s post-Go cue, significant correlation clusters emerged over the left sensorimotor cortex (Fig. 8c bottom, $P$ = [0.028, 0.002, 0.012] uncorrected). Notably, the cluster at 0.1 s occurred before the button was pressed (see RT histogram in Fig. 8c, bottom). The positive correlation (small ERF amplitude, short RT, and vice versa) likely corresponds to preparatory motor activity reflecting a state of readiness. The negative correlation cluster at 0.25 s post-Go was close to the grand mean of RT (0.261 s) and corresponds to the execution of the button press. This interpretation is supported by the negative sign of the correlation: the larger the ERF amplitude, the shorter the RT, and vice versa. The positive correlation cluster at 0.4 s post Go likely reflects a late component of the motor ERF: at 400 ms

post-Go, a small amplitude corresponds to a short RT and vice versa. These clusters over the left sensorimotor cortex were to be expected and provide a sanity check of the correlation analysis pipeline.

### Event-related fields represent event probability density

In a final analysis, we aimed to identify a direct representation of the event PDF in neural activity time-locked to anticipated events. Using the source-level ERF analysis scheme reported above, we performed a source-level correlation between ERFs and fits of the PDF-based model to RT (Methods). This analysis confirmed the key results of the correlation analysis with RT: At 0.25 s after the Go cue, parietal P3b and cerebellar ERFs and motor ERFs at 0.25 s and 0.4 s were correlated with the event PDF (Supplementary Fig. 11). This analysis revealed a representation of the probability distribution of sensory events in time-locked neural activity after an anticipated event.

In sum, the analysis of time-locked data identified significant correlations between ERFs and RT, as well as between ERFs and the PDF-based model of RT. The clusters over the left sensorimotor cortex illustrate how effector motor activity relates to event probability density. The positive correlation clusters at 0.25 s post-Go demonstrate the sensitivity of the parietal P3b and a cerebellar ERF to event probability density in temporal anticipation.

## Discussion

We investigated the neural correlates of temporal anticipation. Participants learned to anticipate the timing of auditory and visual sensory events, which were distributed in time according to exponential and flipped exponential probability density functions (event PDF). Modeling of RT shows that participants estimated the event PDF, but not the canonical hazard rate.

We report three main results on neural data. First, alpha power in the right IPL and pMTG area, low beta band power in the right SPL, and high beta band power in the left sensorimotor cortex represent the event PDF. These three prediction signals occurred before the anticipated sensory event, suggesting functional links to the generation of RT. Second, time-locked activity after the anticipated sensory event was correlated with the event PDF in the right parietal cortex (P300). Third, these results in time-frequency and time-locked neural data were observed in audition and vision. This is consistent with a supra-modal role of the identified cortical areas in the representation of event probability over time.

The representation of elapsed time is fundamental to the estimation of event PDF. Previous work on time span learning (interval timing without probability manipulation) has reported different neural codes for the representation of elapsed time. In these studies, the error in neural temporal estimates commonly follows the scalar property, i.e., the uncertainty in time estimation increases linearly with temporal durations (i.e., temporal blurring). This principle is reflected in premotor[59], posterior parietal[60], striatal[59], and hippocampal[61] activity.

In temporal anticipation, we propose that temporal uncertainty does not follow the scalar property but that it follows event probability (probabilistic blurring): where probability is high, temporal estimates are precise and vice versa. Our results offer neural evidence for this computational principle in different cortical areas; however, the specific underlying neural mechanisms remain to be determined. In rodent striatum and premotor cortex, neural sequences encode time spans, following associative learning rules[59]. Interestingly, these sequences cover the alpha and beta ranges that were also observed in our work. They provide a candidate mechanism for estimating the event PDF and, through associative learning, may implement the probabilistic blurring that our data mandates.

It is widely accepted that the computational processes underlying time estimation are reflected in neural population dynamics[18]. At the (coarse) level of the cortical regions, the representation of time between sensory events is often linked to parietal activity. In macaques, the posterior parietal cortex has been implicated in the representation of time[60]. In humans, activity in the right posterior inferior parietal lobe is related to event timing[62,63], whereas the left parietal lobe does not seem to play such a critical role in the processing of stimulus timing[64,65]. These close connections between basic timing aspects and right parietal areas complement the right lateralization we observe in the processing of event probability over time.

How does the observed lateralization to right parietal areas relate to the generation of a left-lateralized motor command? SPL and IPL are densely connected to a large number of other brain areas[66–68]. The posterior parietal cortex hosts a bimanual representation of the limbs[28], and posterior parietal neurons code for movements of the contralateral limb[29,69]. This connectivity likely supports time-critical motor commands, as required by our task demands.

Temporal anticipation is closely linked to the deployment of attention over time[12]. Both functions critically depend on the estimation of time. However, neural estimates of elapsed time are subjective, i.e., time is associated with uncertainty[11,39,43]. Recent work proposes the right SMG as a locus for the subjective experience of time[70]. Our results identified a representation of event probability over time in the right SMG. In light of the right parietal lobe's association with attention, e.g., to brief visual stimuli[71], auditory selective attention[72], and temporal visual attention in general[64,73], the co-modulation between alpha power and probability-driven RT may be interpreted as a temporal attention phenomenon.

Our analysis identified a modulation of the parietal P300 (P3b) ERF component by event probability density. This was observed in audition and vision, which is in accordance with this component's known independence from input modality[51]. In decision-making tasks involving a choice between options, the P300 occurs before the choice is instantiated and is argued to (partly) represent decision processes[74], such as evidence accumulation[75]. In our experiment, which does not require complex decisions, the P300 occurred *after* the button response, which suggests a role in the later processing stages of event probability density.

Time-locked activity in the cerebellum was correlated with RT and approximated event probability over time. This is in line with the cerebellum's role in planning and execution of motor actions[57] and motor timing[76]. Recently, the cerebellum was linked to other time-based, functional domains, such as attentional modulation of perception[77] and time interval prediction[78]. The cerebellum's vast connectivity to (pre-)motor[79] and prefrontal cortices[80] also includes reciprocal connections to posterior parietal integration areas[81], e.g., to SPL[67]. The close temporal proximity between cerebellar and right posterior parietal signals observed here may reflect a functional role of the cerebellum in temporal anticipation in addition to motor control.

In conclusion, this study reveals three dynamic prediction signals two in the posterior parietal and posterior temporal cortical regions and one in sensorimotor cortex. These signals represent the probability of sensory events over time at the time scale of seconds – a key variable underlying temporal anticipation.

## Methods

The experiments were approved by the Ethics Committee of the University Hospital Frankfurt. Written informed consent was given by all participants prior to the experiment.

### Participants

24 healthy adults (15 female), aged 21–34 years, mean age 27 years, participated in the experiment. All were right-handed and had normal or corrected-to-normal vision, reported no hearing impairment, and no history of neurological disorder. Participants received € 15 per hour. One participant was excluded from the analysis of MEG data because the anatomical MRI data were corrupted and could not be retrieved.

## Experimental task and stimuli

In the MEG booth, participants performed visual and auditory blocks of trials of a Set-Go task. In the task, a SET cue was followed by a Go cue (Fig. 2a). Participants were asked to respond as quickly as possible to the Go cue with a button press (Current Designs Inc., Philadelphia, PA, USA) using their right index finger. Participants were instructed to foveate a central black fixation dot and restrict blinking to the time-span immediately following a button press. In 10% of trials, no Go cue was presented. In these catch trials, participants were asked not to press the button. This small percentage of catch trials was added to avoid possible strong effects of event certainty towards the end of the Go-time span[38]. A small black circle around the central fixation dot was presented onscreen for 200 ms after a button press indicating the end of the trial. The intertrial interval (ITI) was defined by the onset of the small black circle and the Set cue of the following trial. The ITI was drawn randomly from a uniform distribution (range 1.4 to 2.4. s, discretized in steps of 200 ms).

## Visual stimuli

Two simultaneously presented checkerboards served as both Set and Go cues. They were projected (refresh rate 60 Hz) to the back of a gray semi-translucent screen located at a fixed distance of approximately 53 cm from the participants' eyes. Each cue was onscreen for 50 ms. The checkerboards subtended approximately 6.5 × 6.5° visual angle and comprised of 5 × 5 small black and white squares of equal size. The center of the checkerboards was located to the left and right of a central fixation dot at a horizontal distance of approximately 7° visual angle and a vertical distance of 0° visual angle. The black and white pattern of the checkerboards was inverted between Set and Go cues.

## Auditory stimuli

White noise bursts of 50 ms length served as Set and Go cues. Each burst featured an 8 ms cosine ramp at the beginning and end. The bursts were presented at approx. 60 dB SPL above hearing threshold as determined by pure tone audiometry (1 kHz, staircase procedure). All auditory stimuli were output via an RME Fireface UCX interface to a headphone amp (Lake People GT-109) and delivered diotically via a MEG-compatible tube-based system (Eartone Gold 3 A 3 C, Etymotic Research, Elk Grove Village, IL, USA). Visual and auditory stimuli were generated using MatLab version 2017b (The MathWorks, Natick, MA, USA) and the Psychophysics Toolbox[82] version 3 on a Fujitsy Celsius R940 computer running Windows 7 (64 bit).

## Temporal probabilities

The time between Set and Go cues, the Go time, was a random variable, drawn from either an exponential distribution (Eq. 1) with parameter $l = 0.33$ or from its left-right flipped counterpart (tau = 3.0303 for both).

$$f(t) = \frac{1}{l} e^{-\left(\frac{t}{l}\right)} \tag{1}$$

Both distributions were delayed by 0.4 s resulting in a range of Go times from 0.4 s to 1.4 s. Sequential effects were reduced by the constraint that no more than two consecutive trials were allowed the same Go time. The Go time distribution was fixed for a pair of consecutive blocks of trials. Per participant the experiment consisted of four visual and four auditory blocks. A single block consisted of 200 trials, of which 20 did not feature a Go cue (catch trials). Per sensory modality, in two blocks of trials, the Go times were randomly drawn from the exponential distribution, and in the other two blocks, they were randomly drawn from the flipped-exponential distribution as described above.

## Models of reaction time

Hazard-rate-based and PDF-based models of RT were constructed to investigate the effect of the Go time distribution on event anticipation. The presented exponential and flipped exponential Go time distributions are characterized by three functions, the probability density function (PDF), the cumulative distribution function (CDF), and the hazard rate (HR):

$$p\left(t_{go}\right) : PDF\ of\ RT\ as\ a\ function\ of\ 'go'\ time\ t_{go}$$

$$c\left(t_{go}\right) : CDF\ of\ RT\ as\ a\ function\ of\ 'go'\ time\ t_{go} = \int_0^{t_{go}} p(u)du \tag{2}$$

$$h\left(t_{go}\right) : HR\ of\ RT\ as\ a\ function\ of\ 'go'\ time\ t_{go} = \frac{p(t_{go})}{1 - c(t_{go})} \tag{3}$$

## Mirrored temporally blurred hazard rate

To arrive at the temporally blurred HR, each Go time PDF was blurred by a Gaussian uncertainty kernel whose standard deviation linearly increases with elapsed time from a reference time point: $\sigma = \varphi \cdot t$. Here, $t$ is the elapsed time and $\varphi$ is a scale factor by which the standard deviation $\sigma$ of the Gaussian kernel increases. The equations for the corresponding temporally blurred functions are:

$$p_S\left(t_{go}\right) = \frac{1}{\varphi t_{go} \sqrt{2\pi}} \int_{-\infty}^{\infty} p(\tau) \cdot e^{-(\tau - t_{go})^2/(2\varphi^2 \tau^2)} d\tau \tag{4}$$

$$c_S\left(t_{go}\right) = \int_0^{t_{go}} p_S(u)du \tag{5}$$

$$h_S\left(t_{go}\right) = \frac{p_S(t_{go})}{1 - c_S(t_{go})} \tag{6}$$

For a given Go time, $t_{go}$ the PDF is convolved with a Gaussian kernel centered at $t_{go}$ (Eq. 4). At $t_{go} = 0.4 s$ after Set cue onset, the kernel has a standard deviation $\varphi \cdot 0.4$. Similarly at $t_{go} = 1.4 s$ the kernel has standard deviation $\varphi \cdot 1.4$. In the computation of the subjective PDF, the definition of the PDF was extended to the left and right of the Go time range:

$$p_{padded}\left(t_{go}\right) = \begin{cases} 0, (0.4 - 3 \cdot \varphi \cdot 0.4) \leq t < 0.4 \\ p\left(t_{go}\right), 0.4 \leq t \leq 1.4 \\ 0, 1.4 > t \geq (1.4 + 3 \cdot \varphi \cdot 1.4) \end{cases} \tag{7}$$

The extensions were equal to three standard deviations of the Gaussian kernel (encapsulating 99.7% of the Gaussian uncertainty function) at the shortest and longest Go times. Then the integral in Eq. (5) was computed between these new extrema [(0.4 − 3 · φ · 0.4), (1.4 + 3 · φ · 1.4)] instead of the impractical interval of minus to plus infinity. The selection of $\varphi = 0.21$ was consistent with previous research[11,38,39,83,84]. For $\varphi = 0.21$ the temporal range of the extended PDF (Eq. 7) becomes [0.148, 2.28] s which is also the range of integration in the computation of the subjective PDF in Eq. (4). The PDF of each distribution, $p(t_{go})$ was normalized so that its integral from 0.4 s to 1.4 s was 0.9 which reflects the 10% catch trials that did not feature a Go cue. The HR was computed based on the PDF and CDF (Eq. 6). To arrive at the to-be-fit HR variable, the HR was mirrored around its mean

(Eq. 8).

$$x_{mh}\left(t_{go}\right) = -\left(h\left(t_{go}\right)-\bar{h}\right)+\bar{h} = -h\left(t_{go}\right)+2\cdot\bar{h} = -\frac{p\left(t_{go}\right)}{1-c\left(t_{go}\right)}+2\cdot\bar{h} \quad (8)$$

where

$x_{mh}:$ "mirror" of the hazard rate of the PDF

$\bar{h}:$ mean HR

## Reciprocal probabilistically blurred event probability density function

Probabilistic blurring constitutes an alternative hypothesis to the temporal blurring described above. In probabilistic blurring, the uncertainty in elapsed time estimation depends on the probability density function of event occurrence: Go times with a high probability of event occurrence are associated with low uncertainty in time estimation and vice versa, irrespective of the Go time duration[39]. In probabilistic blurring, the standard deviation of the Gaussian kernel scales according to the PDF of event occurrence. In order to use realistic values for the standard deviation of the blurring Gaussian kernel, the minimum and maximum values were set accordingly to the temporal blurring case as

$$\sigma_{\min} = \varphi \cdot t_{\min} = \varphi \cdot 0.4 \text{ and } \sigma_{\max} = \varphi \cdot t_{\max} = \varphi \cdot 1.4 \quad (9)$$

The value of $\varphi$ was likewise set to 0.21. The PDF under investigation was then scaled so that its minimum value is $\sigma_{\min}$ and its maximum value $\sigma_{\max}$.

If $p_{\min}$ and $p_{\max}$ are the minimum and maximum values, respectively of the PDF under investigation, then the function used for computing the standard deviation of the Gaussian kernel based on the PDF $p(t)$ was defined as:

$$s(t) = \left[1 - \frac{(p(t)-p_{\min})}{(p_{\max}-p_{\min})}\right] \cdot (\sigma_{\max}-\sigma_{\min}) + \sigma_{\min} \quad (10)$$

The term inside the brackets demonstrates that when the probability $p(t)$ is low the standard deviation of the Gaussian kernel approaches $\sigma_{\max}$ while when the probability increases, $s(t)$ approaches $\sigma_{\min}$.

Based on Eq. (10) for determining the standard deviation of the Gaussian kernel the probabilistically blurred PDF $p_p(t)$ was computed as:

$$p_p\left(t_{go}\right) = \frac{1}{s(t)\sqrt{2\pi}} \int_{-\infty}^{\infty} p(\tau) \cdot e^{-(\tau-t_{go})^2/(2s(t)^2)} d\tau \quad (11)$$

Finally, in order to implement the Gaussian blurring of Eq. (11) at the extrema of Go-times, the definition of the PDF was extended to the left and right of the actual stimulus presentation interval by three standard deviations of the corresponding smoothing Gaussian kernels, similar to the temporally blurred case described in Eq. (7), as:

$$p_{padded}\left(t_{go}\right) = \begin{cases} 0, (0.4 - 3 \cdot s(0.4)) \le t < 0.4 \\ p\left(t_{go}\right), 0.4 \le t \le 1.4 \\ 0, 1.4 > t \ge (1.4 + 3 \cdot s(1.4)) \end{cases} \quad (12)$$

The extensions of the Go times range depend on the standard deviation function $s(t)$, which itself depends on the probability density

function. To arrive at the to-be-fit PDF variable, the reciprocal of the probabilistically blurred PDF was computed: 1/PDF (Eq. 13).

$$x_{op}\left(t_{go}\right) = \frac{1}{p\left(t_{go}\right)} \quad (13)$$

where

$x_{op}:$ 'reciprocal' PDF

To illustrate the effect of the nonlinear reciprocal computation on the probabilistically blurred PDF, we compared it to the linear mirroring explained above and plotted both the probabilistically blurred PDF in its mirrored and reciprocal version in Fig. 3c.

## Additional models of reaction time

The mirrored probabilistically blurred HR variable was computed. The probabilistically blurred PDF of each distribution was normalized so that its integral from 0.4 s to 1.4 s was 0.9 which reflects the 10% catch trials that did not feature a Go cue. The probabilistically blurred PDF was integrated to arrive at the probabilistically blurred CDF, and the probabilistically blurred HR was computed (Eq. 3). To arrive at the to-be-fit HR variable, the probabilistically blurred HR was mirrored around its mean (Eq. 8). The reciprocal temporally blurred PDF variable was computed. The temporally blurred PDF (Eqs. 4 and 7) of each distribution was calculated. To arrive at the to-be-fit PDF variable, the reciprocal of the temporally blurred PDF was computed: 1/PDF.

## Modeling RT with a linear model

All variables, the mirrored HR and the reciprocal PDF variables, each in temporally blurred and probabilistically blurred versions, were fit to RT data (first aggregated within Go-times within participants, then averaged across participants) using a linear model. An Ordinary Least Squares (OLS) regression was employed for the computation of the regression coefficients using the MatLab (The MathWorks, Natick MA, USA) fit function. Adjusted $R^2$ was used as a measure of goodness-of-fit for comparing the models' relation to RT.

## MEG data acquisition

The neuromagnetic activity was recorded with a 275-channel system (VSM MedTech Omega, Coquitlam, Canada) equipped with axial gradiometers distributed in a helmet across the scalp in a magnetically shielded room at the Brain Imaging Center, Frankfurt. MEG data were recorded continuously with a sampling rate of 1200 Hz. Participants were seated in an upright position and were asked to remain still during blocks of trials. Head position relative to the MEG sensors was controlled and continually monitored during each experimental block using three position indicator coils in the anatomical fiducial locations (left and right pre-auricular points and nasion). Head position was corrected if necessary between blocks using the Fieldtrip toolbox[85]. Electro-cardiogram and vertical and horizontal electrooculogram were also measured at 1200 Hz to identify eye blinks and movements, and the heartbeat in analysis. An eye tracker (Eyelink DM-890, SR Research Ltd.) recorded participants' eye dynamics at a sampling frequency of 500 Hz.

## MEG data preprocessing

Continuous data were down-sampled to 600 Hz. Data were epoched separately with respect to Set and Go cues and to the button press (− 0.5 to 0.72 s). Artifactual epoches and noisy MEG channels were rejected based on visual data inspection using Fieldtrip's visual artifact rejection routines. Based on visual inspection, trials and MEG channels that featured periods of high variance due to, e.g., eyeblinks or excessive movement during the time span of interest were discarded from the analysis. Heartbeat artifacts were removed using

independent component analysis. Drifts in MEG channels were eliminated by high-pass filtering at 0.2 Hz. Muscle activity was eliminated by low-pass filtering at 110 Hz. Finally, only trials with 0.05 s < RT < 0.75 s were used in the analysis of MEG data and in the modeling of RT. The trial selection process left $N$ = 31,962 trials for analysis (3.5% of all trials removed). Data analysis was performed using Matlab versions 2019b and 2022b.

## Magnetic resonance imaging data acquisition
To build participant-specific forward models for MEG source reconstruction, structural magnetic resonance imaging (MRI) scans (T1-weighted) were obtained for all 24 participants at the Brain Imaging Center, Frankfurt. MRI data were acquired on a Siemens 3 T TRIO scanner with a voxel resolution of $1 \times 1 \times 1\,mm^3$ on a $176 \times 256 \times 256$ grid. Vitamin-E capsules were used to identify anatomic landmarks (left and right peri-auricular points and nasion).

## Time frequency analysis on sensor-space data
The time-frequency decomposition was calculated for each subject and each trial through a transformation based on multiplication in the frequency domain with a single taper (Hanning) with a 0.5 Hz resolution from 3 to 60 Hz in steps of 20 ms over $t = [−0.5, 0.75]$ s using the FieldTrip toolbox function ft_freqanalysis with method 'mtmconvol'. The correlation across Go-times between the spectral power and RT was performed at the single-trial level and not at the frame level as in the ERF case. This was due to the highly skewed distribution of the spectral power, which results in a bias across frame average with significantly different numbers of trials (see Supplementary Methods and Supplementary Fig. 12). For each sensor, each frequency and time point, the spectral power was correlated with the corresponding RT (Spearman rank correlation). At the group level, across all four experimental conditions, statistical significance was tested using a cluster-based permutation test on Spearman's rho to identify channel-by-frequency-by-time clusters in which rho differs from zero (cluster-forming $\alpha = 0.05$, 1000 permutations).

## Time frequency analysis on source-space data
The above correlation-based analysis on sensor-space data was also performed at the source-level representation of the time-frequency data. This source space for each participant comprised of the individual's cortical mantle representation, as extracted from a structural MR scan. This source space was fused with a whole-cortex parcellation atlas (AAL atlas[50]) in order to relate findings to known anatomical areas and assist the interpretation of results. Within the frequency ranges of 7–12, 15–22, and 23–30 Hz and within each source, the source-level data were averaged within a time span of 100 ms, centered at t = [−0.35, −0.25, −0.15, −0.05, 0.05, 0.15, 0.25, 0.35] s relative to the Go cue. This range of time windows was selected for all four conditions based on the results of the sensor-level analysis. The inverse solution was derived by the DICS beamformer[49]. For each of the three frequency bands of interest, the cross-spectral density matrix used for the inverse solution was derived by averaging the cross-spectral density matrices across all time points and frequencies of interest.

Then all spectral complex coefficients for all time points (20 ms resolution) and trials were projected through the DICS spatial filters into source space. The power was then computed from the amplitude of each projected complex coefficient. These power values were then averaged within frequency bands-of-interest and time windows of 100 ms, centered at $t = [−0.35, −0.25, −0.15, −0.05, 0.05, 0.15, 0.25, 0.35]$ s relative to the Go cue. This was done for each trial. The average power value per window and frequency band was Spearman correlated with RT across trials.

At the group level, across all four experimental conditions, and for each of the time windows defined above, a cluster-based permutation test on Spearman's rho was performed (cluster-forming $\alpha = 0.05$, 1000 permutations).

An additional correlation analysis investigated the relationship between the *event PDF* and source-level power. At the single-participant level and within each condition, the single-trial RTs were fit with the reciprocal probabilistically blurred event PDF variable as described above. These fits were computed in running batches of $N = 15$ RTs to capture potential fluctuations in average RT over the course of the two experimental blocks of a single condition. In each condition, all batches of fit PDF-based model were concatenated and Spearman correlated with source-level power at the single-trial level within each frequency band-of-interest followed by a cluster-based permutation test as described above. This analysis was performed separately for four windows of 100 ms length, covering the time span of $t = [−0.4, 0\,s]$ relative to the Go cue. Within each frequency band-of-interest, the four resultant correlation clusters were averaged across time for plotting.

## Analysis of eye blink data
Muscle activity related to eye movements was recorded with dedicated electrodes (MEG channels VEOG and HEOG). The single-trial data were locked to the time point of the button press and baselined using a time window of $t = [−0.25, −0.05]$ s relative to the button press. The following within-condition analyses were performed. The average ERP was computed across trials and averaged across participants (Supplementary Figs. 4 & 5). At the single-trial level, the time point at which the eye-blink signal (HEOG) is maximal was computed over $t = [0, 1.75]$ s relative to the button press (Supplementary Figs. 6). At the single-trial level, the time point at which the pupil diameter (measured by eye-tracking) is minimal was computed over $t = [0, 1.75]$ s relative to the button press (Supplementary Fig. 7).

## Event related fields analysis on sensor-space data
Within each condition, the single-participant MEG data was time-locked to the Go cue. The data was aggregated within adjacent pairs of consecutive Go-times (frames): For the first frame, the activity from all trials with Go-times = [0.4, 0.4167] s was averaged; the second frame comprised Go-times = [0.4333, 0.45] s and so forth. This procedure was also applied to the RT. The aggregation reduced the number of unique Go-times from 60 to 30 ( = 30 frames). Before the averaging of the ERF, the data was baselined either to the pre-SET or to the pre-GO period. The pre-SET period ($t = [−0.5, 0]$ s) was selected in the analysis of activity preceding the Go cue, whereas the pre-GO period ($t = [−0.5, 0]$ s) was selected in the analysis of activity following the Go cue. At the single-participant level, for each condition and for each channel-by-time-point duplet, Spearman rank correlation was computed between the 30 frames of MEG data and the 30 averaged RTs. Note that the resultant rho has the same dimensionality as the within-participant grand average of MEG data (channels-by-time-points). At the group level, within each experimental condition, statistical significance was tested using a cluster-based permutation test on Spearman's rho to identify channel-by-time clusters in which rho differs from zero (cluster-forming $\alpha = 0.05$, 1000 permutations).

## Event related fields analysis on source-space data
We used a linearly constrained minimum variance beamformer LCMV[86]. The covariance matrix used for the computation of the spatial filter was derived from the average across trials. The time period used for the computation of the covariance matrix was $t = [−0.5, 0.75]$ s relative to the Go cue. The average, baselined ERF for the computation of the covariance matrix was derived by averaging all trials from all frames so that the spatial filters are common for all frames. The sensor-level average ERF for each frame (see above) was projected to source space through the common spatial filters. For each source location, Spearman correlation was computed between the per-frame averaged ERF and the

corresponding RTs. This was computed for each participant and experimental condition. Rho was averaged across time within windows-of-interest that were chosen based on the sensor-space clusters (30 ms windows centered around 0.25 and 0.4 s post Go) and one additional 30 ms window centered at 0.1 s post Go to probe for early correlations before the button response, all for each participant and experimental condition. At the group level, across all experimental conditions, the non-parametric cluster-based permutation statistics were computed on averaged Spearman's rho to identify clusters of sources in which rho differs from zero (cluster-forming $\alpha = 0.05$, 1000 permutations).

An additional correlation analysis was performed to directly investigate the relationship between the event PDF and source-level ERF. At the single-participant level and within each condition, the single-trial RTs were fit with the reciprocal probabilistically blurred event PDF variable as described above. These fits were computed in running batches of $N = 15$ RTs to capture potential fluctuations in average RT over the course of the two experimental blocks of a single condition. In each condition, all batches of fit PDF-based models were concatenated, and Spearman correlated with the per-frame averaged ERF for each source location for each participant, using the regime described above. At the group level, across all experimental conditions, the non-parametric cluster-based permutation statistics were computed on Spearman's rho to identify clusters of sources in which rho differs from zero (cluster-forming $\alpha = 0.05$, 1000 permutations).

### Visualization of source-space correlation analyses
For visualizing the correlations with source-space spectral power and RT and the source-space event-related fields and RT, we used the cortical surfaces from the Human Connectome Project, and specifically the average cortical surface from the 1200 Subjects Group Average Data[87]. In the HCP, the cortical surfaces of all subjects were first extracted using FreeSurfer and then they were registered to the Conte69 surface-based atlas and the '164k_fs_LR' atlas mesh. In the current study, we employed the subsampled '32k_fs_LR' mesh due to the lower spatial resolution of the MEG modality.

### Reporting summary
Further information on research design is available in the Nature Portfolio Reporting Summary linked to this article.

## Data availability
The reaction time data used in the modeling are provided in this paper (Supplementary Data 1). The magnetoencephalography and and eye-tracking data are currently analyzed in a dedicated project and will be made publicly available upon project completion. The specific data that support the findings of the paper are available from the corresponding author upon request. Source data are provided in this paper.

## Code availability
Code for analyzing the data is available from the corresponding author upon request.

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

## Acknowledgements

We thank Simone Franz and Claudia Lehr for help with data acquisition. Data were provided [in part] by the Human Connectome Project, WU-Minn Consortium (Principal Investigators: David Van Essen and Kamil Ugurbil; 1U54MH091657) funded by the 16 NIH Institutes and Centers that support the NIH Blueprint for Neuroscience Research; and by the McDonnell Center for Systems Neuroscience at Washington University.

## Author contributions

M.G.: Conceptualization, Experimental design, Data acquisition, Formal analysis, Visualization, Writing - original draft, Writing - Review & Editing. D.P.: Conceptualization, Supervision, Writing - original draft, Writing - Review & Editing, Resources. G.M.: Conceptualization, Experimental design, Formal analysis, Supervision, Writing - original draft, Writing - Review & Editing.

## Funding

## Competing interests

The authors declare no competing interests.
