## [Transparent Peer Review file · Nature Communications]

Neural signatures of temporal anticipation in human cortex represent event probability density

Corresponding Author: Dr Matthias Grabenhorst

Version 0:

Reviewer comments:

Reviewer #1

(Remarks to the Author)

Review of Grabenhorst Poeppel

202: "We focused the analysis on the time span preceding the Go cue (Fig. 2a) since this is when a representation of the event PDF should be reflected in neural activity."

Some amplification of the thinking is required here. Why should it occur before the activity triggered by the Go cue, given that it is the velocity of the triggered activity that is altered by knowledge of the pdf. I assume the thought is that the earlier activity is setting up the neural circuitry that will be activated by the Go signal to perform most precisely at an upcoming time.

265: "the correlation between pre-Go-cue spectral power and RT can be interpreted as a neural representation of the event PDF" Would it not be better to say that they are indicative of or a reflection of a neural representation? Does anyone believe that there is a neural process that reads the information encoded in the fluctuations in magnetic field spectral power? I assume that these temporally and spatially localized fluctuations are indicative of relevant neural activity at those spatio-temporal locations.

273 "In each of these two negative clusters, only the significant sensors within the corresponding frequency band were selected." I don't understand this. Are the sensors themselves selectively tuned to specific frequencies? That's what it seems to suggest. Or do you mean to say that the signals that they entered into their next analysis came only from those sensors located in the regions where there was a significant dip in these power bands in anticipation of the Go signals?

412: "The correlation analysis between spectral power and RT reveals that, before anticipated auditory and visual sensory events, alpha, low beta and high beta spectral power dynamics represent an approximation of the event PDF." Make it clearer what the basis of this assertion is. The behaviorally defined pdf is specified with regard to its location in time following (not preceding) the Go signal and by its slash-like shape (Figure 2d&e). The power fluctuations are not in the same temporal location nor do they have the same shape, so one needs to explain why they may be said to approximate the behaviorally estimated pdf.

577: It is argued that the computational processes underlying such time estimation are directly linked to neural population dynamics¹⁸ Not sure what is being asserted here. Suppose for the sake of argument one imagined that the computations were carried out intracellularly by molecular machinery and that the spiking activity that (I assume) is the dominant source of MEG spectral power, transmits the results of those computations to other neurons. Would that constitute a direct linkage? What would constitute indirect linkage?

588: "One obvious question is how the observed lateralization to right parietal areas relates to the generation of a left lateralized motor command, as required by our task." The lengthy musings that follow the posing of this question and that reappear in the Discussion serve only to emphasize how conjectural and vague are the possible inferences about areal functions that one might draw from these findings. There is nothing to be ashamed of about this because even the current story about the function of, for example, even V1 is incomplete and vague at many points, and for that region we have vastly more and more direct data on neural activity under controlled conditions. However, the tenuous and vagueness of these musings does call into question the value of including them in the publication of the findings.

The writing is inelegant. A technically sophisticated copy editor could make it better. Among the simplest edits would be to strike out the emphatic adverbs and adjectives: "Remarkably" "convincing" etc. Beyond that, simplification of sentence structure would improve the exposition.

Here for illustration is one of many awkward run-on sentences: 438 "This analysis further demonstrates that neural dynamics in alpha and beta frequency bands represent the event probability distribution over time prior to anticipated sensory events in higher-order cortex and in sensorimotor cortex."

I recommend radically reducing the speculative remarks about functionality and dialing back—or clarifying—the claim that the spatio-temporally localized fluctuations at certain frequencies of the spectral power approximate the behaviorally estimated pdf. To my mind, what these fluctuations indicate is simply where to look for more direct data on how the pdf is derived and how it controls the reaction time. The complexity of the correlational analyses and the multitude of variables that come into play make it hard to extract a clear take home message.

C. R. Gallistel

Reviewer #2

(Remarks to the Author)

This paper uses an implicit timing task (i.e., one in which the task itself—pressing a button in response to a stimulus—does not rely on explicit temporal information). The standard view is that during implicit timing tasks subjects create a temporal expectation of when the stimulus will arrive based on the hazard function (the probability an event is about occur given that it has not yet occurred). Here the authors provide compelling evidence that temporal expectation is not based on the hazard rate, but the PDF (probability density function, i.e., the distribution of the experienced intervals). The authors show that reaction times (RTs) are not quantitatively (or in some cases qualitatively) captured by the hazard rate, but well captured by the PDF. MEG studies demonstrated a supramodal inverse correlation between RT and alpha power in the R superior parietal cortex, and a correlation between beta power and RT in the L sensorimotor cortex. The paper significantly advances, and alters, our current understanding of this form of implicit timing, and provides significant insights into the potential brain areas contributing to timing and the encoding of the PDF.

My major comments are:

1. It was not clear why the PDF model used the probabilistic blurring rather than temporal blurring as in the hazard rate model. Is the probabilistic blurring supposed to capture the perceptual learning of the intervals (which generally does not happen quickly)? As it currently stands, the reader can fairly ask how much of the difference between the models stems from the use of temporal blurring in the hazard mode versus probabilistic blurring in the PDF model? It seems necessary to perform an apples-to-apples comparison of both models using the same blurring function—either the hazard with the probabilistic blurring or the PDF with the temporal blurring. As I understand it, using the same blurring function in both models should not change the results that much?

2. The abstract states "These results show that supra-modal representations of probability density across cortex underlie the anticipation of future events". To arrive at the conclusion that the observed MEG signals represent the PDF wouldn't it be necessary to directly contrast the alpha and beta dynamics during the exponential and flipped distributions? Relatedly, can the alpha and beta dynamics not be encoding the PDF but elapsed time itself—that is, can elapsed time from the Set cue be decoded from changes in alpha/beta power?

Additional comments and suggestions:

While the correlation values between alpha/beta power and RT are significant they seem fairly weak. Is this normal in these types of MEG analyses? Is it possible to perform correlations between the phase of alpha and RTs (e.g., Lakatos, ..., Schroeder, 2008).

Figure 2 provides a nice analyses of the RT results with the exponential and inverse exponential PDFs. However, RT is often inversely correlated with accuracy. Was there a significant number of false (early) responses? If so where there any differences in accuracy (as measured by false response) between modality and distribution, and on the timing of these false responses between the distributions?

In the Discussion it would be helpful to better place the current results in the context of existing models and the large amount of data on timing from animal studies. Particularly if it turns out that one can decode time from the alpha/beta dynamics. A key point of the paper is the PDF is simpler to calculate and thus more biologically plausible. Indeed, it has been pointed out that the orthogonality of neural sequences are well suited to read out time using associative learning rules (Zhou & Buonomano, 2020), providing a simple mechanism to capture the PDF.

I think Figure 2f-g is referenced after figure 3. Maybe Figure 3 should be 2 (since it seems to be explaining the methods used in current Figure 2)?

Why is the schematic of the correlation shown in Fig. 4C, if the correlation data is in Figure 5?

Line 99. Provide the tau's here (as well as in the methods).

Line 204. "respectively" is not needed.

Results. Provide the df's with the stats results.

Reviewer #3

(Remarks to the Author)

The present work aims to identify the cortical representation of temporal statistics. Human MEG and reaction time data are acquired in the context of a multimodal (auditory/visual) cued paradigm. A large arsenal of data analysis strategies specific to MEEG data, such as time and time-frequency analysis on sensor and source space, are pursued in the quest to find the cortical representation of temporal statistics. Unfortunately, the results are inconclusive and echo findings observed in different experimental contexts without manipulation of temporal expectation. The exploratory nature of the analyses is acceptable when clearly stated and introduced, but the introduction does not justify the extensive application of various analyses to find "the cortical representation of temporal statistics." The present work will benefit from structuring and shortening the analyses with respect to the actual hypothesis.

The assertion that the probability of occurrence of a future event is represented in a cortical region is a misconception (p. 2, lines 51-55). Probability over time cannot have a cortical representation.

Figure 1a claims neural systems count events over time. This hypothesis is unfalsifiable and appears fictional.

The central question already assumes the existence of a cortical representation of event probability (p. 2, line 59). Consequently, the logical next step is to identify the cortical representation of temporal statistics (p. 3, line 76). This is a circular logic and the premise seems questionable. This representation shaping motor responses is overly abstract.

In contrast, the hypothesis that a high likelihood of an event leads to faster reactions (p. 3, line 83) is trivial and empirically derived.

The statement that the human brain represents event probability over time (p. 7, line 177) is speculative and not a factual basis for the proposed modeling work.

The preset cue baseline interval (p. 8, line 207) needs adjustment to avoid post-cue data contamination. Given a sliding window of 2 sec (0.5Hz resolution), it is unclear how the peri-button press activity is smeared in time.

There is an inconsistency in the units of relative power in TFR and topographies (p. 9, Fig. 4).
p.9 line 211: "Activity related to Go cue anticipation based on task demands" is a well-established beta decrease in the motor cortex prior to a button press. Task demands are minimal beyond the instruction to press a button upon the go signal.

The statement that alpha suppression dynamics are consistent with an anticipation signal (p. 9, lines 240-241). Alternative explanation for this rather speculative conclusion is the fact that eye blinks are consistently performed with the button press. The authors could check using the VEOG and the HEOG channels and compute the ERP. Such analysis will likely yield a consistent blink time course in VEOG and less so in the HEOG. Accordingly, alpha power is changed in preparation and under closed eyelids. As such, this has little to do with the anticipation.

The authors need to clarify if the observed peaks reflect within-cluster separation or harmonics (p. 10, lines 269-271).

The method used does not justify ROI selection (p. 10, lines 272-274). The claim appears cherry-picked and circular without identifying a „significant cluster“ considering all possible locations (https://www.fieldtriptoolbox.org/faq/how_not_to_interpret_results_from_a_cluster-based_permutation_test/).

The statement that mu is not modulated by anticipation due to a lack of significant correlation (p. 11, lines 291-293) is incorrect. It merely indicates no statement can be made regarding the hypothesis.

The negative correlation within the first 300 ms post-cue (p. 11, lines 295-297) might be associated with evoked activity. Concluding this effect reflects ongoing oscillation and functional implementation in temporal anticipation is premature.

The spectral decomposition of evoked activity likely contributes to the observed frequency band (p. 12, lines 302-303).

The relationship between RT and alpha-beta modulations before button press (p. 12, lines 314-318) replicates earlier work and is commonly accepted. The conclusion that this relates to prediction and temporal anticipation is premature.

The conclusion that peri-button press changes in spectral power represent the neural representation of event PDF underlying anticipation (p. 12, lines 319-322) is too strong and unsupported by the data.

The frequency resolution suggested by DICS method results (p. 13, line 340) appears inconsistent. A window of 100 ms results in Δf of 10 Hz. How were 7,8,9 Hz etc. estimated?

The section discussing significant clusters is redundant and circular (p. 13). The authors already rejected the H0 on sensor level and yet using the same data the hypothesis is used again to evaluate effects on source level. Acknowledging this fact and avoiding overstatement is recommended (i.e. "After 0.15 s post cue, there were no significant clusters.")

Similar sensor and source maps have been reported in other contexts, such as the HCP WM task (p. 13-15). For example <https://pubmed.ncbi.nlm.nih.gov/29803957/>. The right lateralization of the effects appears not to be specific to the "representation of event PDF" or the "supra-modal functional role of SPL in temporal anticipation." In fact, it appears that the authors could perform the same analyses on publicly available data and arrive at a similar conclusion, despite the fact that event anticipation, temporal predictions, and sensory modality were not manipulated.

The analyses on p. 18 are also rather circular. The identification of temporal clusters and their use as TOI is not justified by the methodology.

Version 1:

Reviewer comments:

Reviewer #1

(Remarks to the Author)

Line 219: "...that neural activity before an anticipated cue may prepare the neural circuitry that will be activated by the cue...."
I suggest that 'potentiate' is better (more specific) than 'prepare'

Line 441: 'before to' ungrammatical

p. 584-586: This sentence treats temporal probability as an alternative to the well established scalar variability in the estimation of elapsed intervals. While it may be possible to elaborate a model in which this is the case, they do not do so here, and, unless my intuitions are leading me seriously astray, such a model would have to make some pretty strong assumptions, which would seem to me highly implausible. The subject's measure of the time elapsed presumably provides the support vector for the subject's probability density function in these experiments, where they imposed two different (mirror imaged exponential) probability density functions on the elapsed time at which the Go signal would occur. My own conception, which I had assumed was the authors' conception, is that the subjects impose corresponding response-potential functions on their measure of the interval elapsed after the Set signal. Thus, when the Go-probability steps up to its maximum at 0.4 s after the Set signal, the response to the Go signal is maximally potentiated and therefore occurs with the minimum latency when the subject's measure of time elapsed reaches 0.4s. When the Go-probability steps up to a minimal probability at 0.4s and then accelerates to a maximum at 1.4s, the subject's response-potential function has the same profile, but again, this profile is supported by the subject's measure of the elapsed interval. I haven't thought this out carefully but if my intuitions are right, then the subject's function will be the convolution of the experimenter-imposed exponential or flipped exponential and the subject's Gaussian distribution on elapsed time, which is parameterized by its mean, which equals the actual time elapsed, and a constant(!) cov, hence an uncertainty (variability) proportionate to elapsed time (scalar variability). Intuitively, the horizontal variability in the support vector must spread out the probability density function. For a model to claim that the only factor is the pdf on time, it would have to assume that variability in the support vector did not increase over time, which seems to me implausible. More importantly, there is a lot of experimental evidence against

Reviewer #2

(Remarks to the Author)

The authors have done an excellent job in addressing my concerns. In particular it is very helpful to know that it was not possible to decode elapsed time from alpha/beta, and that there was no information in the phase. It is also now much clearer that the correlations were performed across the four relevant conditions, making the reported findings of correlates of the PDF in the IPL, PMTG important and compelling.

Overall the authors have gone above and beyond in addressing the concerns of all reviewers concerns.

Reviewer #3

(Remarks to the Author)

I was wrong. I have been taught a lesson on various levels.

I appreciate the clarity, depth, and structure of the response letter. I follow the author's arguments. I have realized that without providing a context, my critique achieved nothing but appeared cynical at best. This wasn't my intention, and I apologize to the authors.

I was coming from a standpoint of a general problem in the field assigning various sorts of psychological functions to often similar, sometimes the same voxels, brain rhythms, and time points. Doing so, I put words in the author's narrative, which is naively foolish. Pursuing research within the accepted premises and current standards in the field, as the present work clearly does, is the scientific method and cannot be criticized without context. Which I have not provided. The review process is, however, neither the forum nor the format to pursue this debate further.

I appreciate the additional work RE EOG and blinks. My point was that blinks might be consistent with the button press, not zero-phase locked to the button press. In my reading, the supplemental figure demonstrates this. Their execution might be associated with similar preparatory alpha/beta as is the case for button presses. If the latter is correlated with blinks as it appears to be, one would also expect the correlation between alpha and RT. One could median split the trials concerning the EOG blink amplitude (or some sort of percentiles, even trials where no blink was executed after the button press maybe). In this case, the relationship reported in Figure 5c shouldn't change. If it does, however, I feel that the conclusion that the correlation is not driven by (a preparation for) eye blink activity entailed in the motor act button press could be revisited. I do not suggest the authors should do this in the present work. Just the context for the point I had raised. Additional revision rounds appear, in my view, not necessary. I recommend publication. Thank you.

We thank the reviewers for their thorough and thoughtful comments. The reviewers raised a range of relevant issues that we have addressed with an extensive revision of the manuscript including new analyses of MEG data, eye-blink data and reaction times. In the following, we respond in detail, interleaving our replies in blue.

Reviewer #1

Comment 1

Reviewer: 202: “We focused the analysis on the time span preceding the Go cue (Fig. 2a) since this is when a representation of the event PDF should be reflected in neural activity.”

Some amplification of the thinking is required here. Why should it occur before the activity triggered by the Go cue, given that it is the velocity of the triggered activity that is altered by knowledge of the pdf. I assume the thought is that the earlier activity is setting up the neural circuitry that will be activated by the Go signal to perform most precisely at an upcoming time.

Reply: The reviewer suggests to better explain why we selected the pre-Go period as a time-span-of-interest. We agree and added a statement that explains our rationale (l. 218-220).

Comment 2

Reviewer: 265: “the correlation between pre-Go-cue spectral power and RT can be interpreted as a neural representation of the event PDF” Would it not be better to say that they are indicative of or a reflection of a neural representation? Does anyone believe that there is a neural process that reads the information encoded in the fluctuations in magnetic field spectral power? I assume that these temporally and spatially localized fluctuations are indicative of relevant neural activity at those spatio-temporal locations.

Reply: Agreed. We corrected the statement (l. 276-278).

Comment 3

Reviewer: 273 “In each of these two negative clusters, only the significant sensors within the corresponding frequency band were selected.” I don’t understand this. Are the sensors themselves selectively tuned to specific frequencies? That’s what it seems to suggest. Or do you mean to say that the signals that they entered into their next analysis came only from those sensors located in the regions where there was a significant dip in these power bands in anticipation of the Go signals?

Reply: We apologize for not being clear on this point. What we meant is that we selected the sets of sensors (alpha and low beta) for plotting (not for a next analysis), see Fig. 5. We changed the sentence (l. 285-287).

Comment 4

Reviewer: 412: “The correlation analysis between spectral power and RT reveals that, before anticipated auditory and visual sensory events, alpha, low beta and high beta spectral power dynamics represent an approximation of the event PDF.”

Make it clearer what the basis of this assertion is. The behaviorally defined pdf is specified with regard to its location in time following (not preceding) the Go signal and by its slash-like

shape (Figure 2d&e). The power fluctuations are not in the same temporal location nor do they have the same shape, so one needs to explain why they may be said to approximate the behaviorally estimated pdf.

Reply: We realized that the relationship between the event PDF and the power modulation requires more explanation at this crucial point in the paper. We added a paragraph that explains how the reaction times relate to the event PDF and how this is connected to the observed power modulation. This paragraph also briefly lays out why the power modulation is considered a prediction signal (l. 435-442).

Comment 5

Reviewer: 577: It is argued that the computational processes underlying such time estimation are directly linked to neural population dynamics¹⁸

Not sure what is being asserted here. Suppose for the sake of argument one imagined that the computations were carried out intracellularly by molecular machinery and that the spiking activity that (I assume) is the dominant source of MEG spectral power, transmits the results of those computations to other neurons. Would that constitute a direct linkage? What would constitute indirect linkage?

Reply: We agree with the reviewer and realize that our statement was not clear. Especially, the wording "directly linked" is confusing. What we intended to say was that, in the literature, the computational processes underlying time estimation are thought to be reflected in neural population dynamics. We introduce this general statement in order to discuss the lateralization of the correlation signals at an anatomical level appropriate for MEG. In the revised manuscript, we changed the sentence accordingly (l. 593-594) and shortened the discussion of this point (l. 593-600).

Comment 6

Reviewer: 588: "One obvious question is how the observed lateralization to right parietal areas relates to the generation of a left lateralized motor command, as required by our task." The lengthy musings that follow the posing of this question and that reappear in the Discussion serve only to emphasize how conjectural and vague are the possible inferences about areal functions that one might draw from these findings. There is nothing to be ashamed of about this because even the current story about the function of, for example, even V1 is incomplete and vague at many points, and for that region we have vastly more and more direct data on neural activity under controlled conditions. However, the tenuous and vagueness of these musings does call into question the value of including them in the publication of the findings.

Reply: We agree and reduced the length of the Discussion (2008 words to 870 words). We briefly discuss four points that are of interest to the timing community: neural representation of probability over time, lateralization of PDF representation, alpha power as a temporal attention signal, event-related fields in temporal anticipation.

Comment 7

Reviewer: The writing is inelegant. A technically sophisticated copy editor could make it better. Among the simplest edits would be to strike out the emphatic adverbs and adjectives: "Remarkably" "convincing" etc. Beyond that, simplification of sentence structure would improve the exposition.

Reply: We removed emphatic adverbs and adjectives and simplified the sentence structure in many instances. We revised the entire manuscript with the reviewer's suggestions in mind.

Comment 8

Reviewer: I recommend radically reducing the speculative remarks about functionality and dialing back—or clarifying—the claim that the spatio-temporally localized fluctuations at certain frequencies of the spectral power approximate the behaviorally estimated pdf. To my mind, what these fluctuations indicate is simply where to look for more direct data on how the pdf is derived and how it controls the reaction time. The complexity of the correlational analyses and the multitude of variables that come into play make it hard to extract a clear take home message.

Reply: We agree and reduced the remarks about functionality (also see our response to Comment 6). We made an effort to better explain why the modulations of spectral power represent the event PDF (l. 435-442, see our response to Comment 4). We say that we identified cortical areas where the probabilistically blurred PDF is represented and point out that the underlying computational mechanisms remain to be determined (l. 586-588). We discuss how neural activity may represent probability over time and relate this to a candidate neural mechanism that may estimate the event PDF (l. 578-592). Lastly, we edited the key sections of the manuscript in order to offer clearer take-home messages (l. 95-98, l. 565-573, l. 631-634).

Reviewer #2

This paper uses an implicit timing task (i.e., one in which the task itself—pressing a button in response to a stimulus—does not rely on explicit temporal information). The standard view is that during implicit timing tasks subjects create a temporal expectation of when the stimulus will arrive based on the hazard function (the probability an event is about occur given that it has not yet occurred). Here the authors provide compelling evidence that temporal expectation is not based on the hazard rate, but the PDF (probability density function, i.e., the distribution of the experienced intervals). The authors show that reaction times (RTs) are not quantitatively (or in some cases qualitatively) captured by the hazard rate, but well captured by the PDF. MEG studies demonstrated a supramodal inverse correlation between RT and alpha power in the R superior parietal cortex, and a correlation between beta power and RT in the L sensorimotor cortex. The paper significantly advances, and alters, our current understanding of this form of implicit timing, and provides significant insights into the potential brain areas contributing to timing and the encoding of the PDF.

Reply: We thank the reviewer for recognizing our efforts to advance our understanding of implicit timing and its neural underpinnings.

Major comment 1

Reviewer: It was not clear why the PDF model used the probabilistic blurring rather than temporal blurring as in the hazard rate model. Is the probabilistic blurring supposed to capture the perceptual learning of the intervals (which generally does not happen quickly)?

As it currently stands, the reader can fairly ask how much of the difference between the models stems from the use of temporal blurring in the hazard mode versus probabilistic blurring in the PDF model?

It seems necessary to perform an apples-to-apples comparison of both models using the same blurring function—either the hazard with the probabilistic blurring or the PDF with the temporal blurring. As I understand it, using the same blurring function in both models should not change the results that much?

Reply: The reviewer asks whether probabilistic blurring is supposed to capture interval learning. The reviewer points out that it is unclear why the hazard rate and PDF models use different kinds of blurring functions (temporal vs. probabilistic). The reviewer suggests to fit additional models to RT (probabilistically blurred PDF and temporally blurred HR) so that the reader can better evaluate the effect of the different blurring hypotheses on the model fits.

In answer to the first question: We think of probabilistic blurring as a function that allows the neural system to fine-tune its precision in time estimation according to its probability estimates: Where event probability is large, temporal uncertainty is small and vice versa. We develop this idea in a previous paper (Grabenhorst, Michalareas, et al. Nat Commun 2019). Probabilistic blurring can be related to interval learning in the sense that it reflects interval *counting* in the build-up of an estimate of the event PDF. We are currently working towards a model that describes this trial-by-trial process of event counting over time which modulates precision in time estimation.

We address the viewer's modeling suggestion with new analyses and explanations in the manuscript. In our previous paper (Grabenhorst, Michalareas, et al. Nat Commun 2019), we perform extensive model comparisons between temporally and probabilistically blurred models that the reviewer asks for. We use this opportunity to replicate these previous modeling results.

We fit the models suggested by the reviewer (Suppl. Fig. 3) and conclude that the probabilistically blurred PDF-based model offers the best fit of all temporally and probabilistically blurred HR and PDF models. We also added a statement to the manuscript that - irrespective of the type of blurring - the PDF-based models outperform the HR-based models (l. 204-209).

Major comment 2

Reviewer: The abstract states “These results show that supra-modal representations of probability density across cortex underlie the anticipation of future events”. To arrive at the conclusion that the observed MEG signals represent the PDF wouldn't it be necessary to directly contrast the alpha and beta dynamics during the exponential and flipped distributions?

Reply: The reviewer is right: in order to claim that the PDF is represented supra-modally in the alpha and beta band signals, one needs to test across the two Go time distribution conditions and across the two sensory modalities. We did exactly that in our analysis: *Within* all four

experimental conditions, Spearman correlation was computed between single-trial spectral power and single-trial RT for each time-frequency-sensor triplet. Across all four conditions, a cluster-based permutation test was run on Spearman's rho in time-frequency-sensor space. This test identifies positive and negative correlation clusters. Crucially, a single cluster can only accommodate positive or negative correlation values. Thus the test identifies clusters of significant correlation in time-frequency-sensor space that are shared by exponential and flipped exponential conditions and also shared by audition and vision. We illustrate this in the following Fig. R1:

Fig. R1 | Illustration of PDF-based regressors on neural data and possible correlation cluster signs.

We realized that this crucial information on the analysis strategy required a better explanation. We changed the section of the manuscript accordingly (l. 268-272).

Reviewer: Relatedly, might the alpha and beta dynamics not be encoding the PDF but elapsed time itself?

Reply: We pondered the same question early on in the project and addressed it by developing an orthogonal experimental design. We do not think that the alpha and beta dynamics represent elapsed time but that they represent the event PDF and lay out our rationale in the following.

Elapsed time numerically increases between Set and Go cues (Fig. R2a left). The flipped exponential PDF also numerically increases between Set and Go cues, i.e. over time (Fig. R2a right). Accordingly, when using either one of these regressors on neural data, it is not possible to determine whether the identified neural activity encodes elapsed time or the flipped exponential PDF. However, in the *exponential* condition, the PDF *decreases* over time (Fig. R2a, middle). So our design is orthogonal with respect to the sign of the slope of the regressors.

In the manuscript, the high beta band dynamics are *positively* correlated with RT (and also with the fitted PDF-based model) in audition and vision and in exponential and flipped exponential conditions. Since time is a monotonically increasing regressor, it may be positively correlated with the neural data in the flipped exponential condition, but it cannot also be positively correlated with the neural data in the exponential condition (see Fig. R2b for illustration).

The same logic applies in the case of the alpha and the low beta band dynamics where we observed a negative correlation (see Fig. R2c for illustration). This demonstrates that the correlation clusters (computed across the four experimental conditions as described above) reflect the PDF but not elapsed time.

Fig. R2 | Illustration of elapsed time and PDF-based regressors on neural data.

Reviewer: Can elapsed time from the Set cue be decoded from changes in alpha/beta power?

Reply: We think that - using the data from the current experiment - it is not possible to decode elapsed time in a meaningful way from the alpha and beta dynamics, because time and probability over time cannot be disentangled. We describe our rationale in the following.

Work on the decoding of elapsed time from neural activity uses targeted experiments, often without the manipulation of probability over time and presenting only a few time points (e.g. Leon & Shadlen, 2003). For example, in the paper that the reviewer refers to in minor comment 4 (Zhou & Buonomano, 2020), only 2 time spans are presented without probability manipulation, i.e. both are equi-probable. In this paper, the data are analysed with respect to the two time points and a clear neural signature is demonstrated at both time points. In contrast, in our experimental design, the PDFs are defined over many more Go times ($N = 60$, $\Delta t = 16.7$ ms) over the time span of 1 second ($t = [0.4, 1.4]$ s), see Fig. R3).

Fig. R3 | Illustration event PDFs defined at 60 Go times over $t = [0.4, 1.4]$ s.

Thus in the decoding of time from neural activity, elapsed time would be a monotonically increasing regressor defined at 60 time points. Crucially, the individual Go times differ in their number of occurrences in the experiment. This yields several conceptual problems when trying to analyse the neural data with respect to elapsed time itself:

1) **Within-distribution analysis.** For example, in the hypothetical case of two different neural signatures, one at $t = 0.4$ s, the other at $t = 1.4$ s, one cannot argue that elapsed time drives the putative differences because – *within-distribution* – the two Go times differ strongly in the number of occurrences ($N = 1$ vs. $N = 19$). This difference likely affects any putative differences in neural signatures. However, when trying to investigate neural correlates of elapsed time, these differences cannot be regressed out, for example, by using the number of occurrences to perform a weighted fit to neural data. Such a weighted fit makes the implicit assumption that the system has learned and remembered the number of occurrences of each individual Go time. This is a separate hypothesis that needs to be tested.

Relatedly, in single-trial analysis, there may be more noise in a neural signal at low probability Go times ($N = [1,2,3\dots]$ occurrences) vs. high-probability Go times ($N = \dots 17,18,19$ occurrences), likely biasing decoding accuracy. It is also not feasible to perform fits on aggregated spectral power for reasons explained in a simulation in Supplementary Methods ("A note on correlation bias", Suppl. Fig. 10).

2) **Across-distribution analysis.** It is not meaningful to investigate the encoding of elapsed time *across-distributions*. If one selected $t = 0.4$ s in the exponential condition and contrasted it with $t = 1.4$ s in the flipped exponential condition, the number of Go time occurrences are identical ($N = 19$). However, any hypothetical difference in neural activity at these time points may not necessarily be driven by elapsed time itself. Instead, it may be driven by differences in other properties of the probability distributions: e.g. at Go time $t = 0.4$ s, the exponential condition has a steep onset of the PDF, whereas at Go time $t = 1.4$ s the flipped exponential condition has a steep offset of the PDF.

3) **Analysis based on binned Go time.** Binning of Go times does not solve the problems of the within-distribution differences in number of occurrences or the across-distribution differences in probabilistic structure. Suppose one assigned e.g. the first 5 Go times (exponential condition) to one bin and the last 5 Go times (flipped exponential condition) to another bin. The contrast between the two bins would be confounded by the differences in PDF slopes within the bins: decreasing PDF slope (bin 1) and increasing PDF slope (bin 2). etc.

We conclude that, in our design, probability and elapsed time are inextricably entwined.

There is also a more general, conceptual problem. One may decide to ignore the distributions and just use elapsed time as a regressor on neural data. In this case, it will be hard to say what the identified neural activity actually reflects. While such a correlation might be related to elapsed time encoding, it cannot be ruled out that it reflects any other function that might change monotonically over Go time, e.g. an attention signal or a value signal.

Taken together, we do not see a clean way to separate time from probability in the current experiment. This should better be the topic of a dedicated paper that uses a targeted experiment to investigate the effect of elapsed time itself on the alpha and beta dynamics in temporal anticipation.

Minor comment 1

Reviewer: While the correlation values between alpha/beta power and RT are significant they seem fairly weak. Is this normal in these types of MEG analyses?

Reply: The type of analysis we used is a correlation computed on within-participant, within-condition data: single-trial spectral power was correlated with single-trial RT (or with single-trial fits of the PDF-based model of RT). Correlation values in single-trial analyses are commonly small.

In general, trial-by-trial variability in alpha and beta band spectral power is large before a sensory cue and it is strongly reduced after a sensory cue (e.g. Daniel et al. Sci Rep, 2019: <https://www.nature.com/articles/s41598-019-53270-7>). Our time span of interest is located before the sensory stimulus (Go cue), where trial-by-trial variability is large and small correlation values are to be expected.

In contrast, in the correlation analyses on the event-related fields (Fig. 8), Spearman's rho is numerically larger, as the sensory ERPs are strong signals. The larger correlation values in the ERP analysis can also (partially) be attributed to a reduction of noise due to the within-frame (= 2 Go times) averaging of the MEG data.

On a related note, we performed data simulations that illustrate why we used these different strategies for the correlations (single-trial data for *time-frequency analyses*, within-frame aggregation for *event-related field analyses*). These simulations show that an aggregation of the time-frequency data leads to a monotonic bias in the correlation coefficient that is due to the distribution of spectral power. The simulations therefore mandate the use of a single-trial analysis strategy ("A note on correlation bias", Suppl. Fig. 10).

Minor comment 2

Reviewer: Is it possible to perform correlations between the phase of alpha and RTs (e.g., Lakatos, ..., Schroeder, 2008).

Reply: The reviewer likely has in mind that this analysis may provide evidence for a mechanism of attentional selection as in Lakatos, et al. Science 2008 who analysed pre-stimulus delta phase and reaction times.

Mechanistically, the rationale presupposes an alignment of alpha phase and anticipation affecting RT. Although it is not known what aspect of phase corresponds to anticipation. A possible way to investigate this is to look at the alpha phase before the GO cue, similar to Lakatos et al. If there is a pattern in the phase angle at this crucial time point in the task, there may be a systematic relationship to RT.

We computed the alpha phase angle over $t = [-0.1, 0]$ (pre Go cue) on the complex Fourier coefficients using the MEG channels of the negative correlation cluster in the alpha band (7–12 Hz) in all conditions. As can be seen in the following plot, alpha phase is uniformly distributed, so there is no evidence for rhythmic sampling in the neural data before the Go cue.

Fig. R4 | Histogram of phase angles computed over $t = [-0.1, 0]$ relative to Go cue.

Lakatos et al. used a dedicated design where the stimuli were delivered with random stimulus onset asynchronies (SOAs) varying between 500 and 800 ms (Gaussian distribution), with a mean of 650 ms which corresponds to the delta frequency (1.5 Hz). In our experiment, we found no evidence for a systematic relationship between alpha phase and RT. This may be due the fact that the Go time intervals are quite long ($t_{Go} = [0.4, 1.4]$ s), accommodating up to approx. 10 alpha cycles, whereas Lakatos et al. used a temporal stimulus structure corresponding to one delta cycle. Also the range of possible Go times was much larger in our experiment than in Lakatos et al.'s experiment (1 s vs. 0.3 s). It may be that the investigation of alpha phase requires shorter and less variable time spans than the ones we investigate here.

Still, we think that the reviewer's idea is very interesting and the literature suggests that there is evidence that the phase of alpha can influence perception (van Rullen TICS 2016: <https://pubmed.ncbi.nlm.nih.gov/27567317/>). But for the reasons outlines above, the investigation of alpha phase in temporal anticipation requires a more targeted experiment.

Minor comment 3

Reviewer: Figure 2 provides a nice analyses of the RT results with the exponential and inverse exponential PDFs. However, RT is often inversely correlated with accuracy.

Was there a significantly number of false (early) responses? If so where there any differences in accuracy (as measured by false response) between modality and distribution, and on the timing of these false responses between the distributions?

Reply: The reviewer asks whether there was a significant number of early responses and whether these numbers differ across modality (aud and vis) and distribution (exp and flip). The reviewer further asks about the precise timing of these false responses.

We addressed these points with new RT analyses. Across the four experimental conditions, there was an average of 1.59% false (early) responses before the Go cue (515 out of 32477 trials, n.b. $32477 - 515 = 31962$ trials, as depicted in Fig. 2c). The number of early responses did not differ significantly between modalities (aud/vis) or between distribution (exp/flip) (Suppl. Tables S1 and S2). We added these results to the manuscript (l. 116-119).

For early responses (button pressed before Go cue), we did not record the ("negative") time between the button press and Go cue onset. The reason for this is that we tuned the system for maximal reliability and accuracy of reaction time measurement after the Go cue. Therefore, we unfortunately cannot investigate the timing of the false responses in this experiment. However, given the small number of early responses (5 to 6 out of 360 per participant) and in light of no significant across-condition differences, this analysis might have been of limited insight.

We further analysed the early responses with respect to go times. Early responses were rare (Suppl. Table S1) and were mostly observed at high probability Go times, i.e. at shorter Go times in the exponential conditions and at longer Go times in the flipped exponential conditions (Suppl. Fig. 2). This pattern was more pronounced in the flipped exponential conditions. This makes intuitive sense: In the flipped exponential condition, Go cues were more likely to occur at the upper bound of the go times range. So participants strongly anticipated a Go cue at longer Go times based on the knowledge that an event will likely be imminent. Still this lead to early responses in only very few trials (1.44% trials in the auditory flip condition and to 1.35% of trials in the visual flip condition).

We further followed the reviewer's line of thought. We looked into the precision of participants' responses and investigated RT variance which is another measure of behavioral accuracy. In simple RT tasks, such as the one we use here, RT variance positively correlates with average RT (Response Times, Luce, 1986). This linear relationship is also observed here in all experimental conditions (l. 114-115, Suppl. Fig. 1).

Minor comment 4

Reviewer: In the Discussion it would be helpful to better place the current results in the context of existing models and the large amount of data on timing from animal studies. Particularly if it turns out that one can decode time from the alpha/beta dynamics. A key point of the paper is the PDF is simpler to calculate and thus more biologically plausible. Indeed, it has been pointed out that the orthogonality of neural sequences are well suited to read out time using associative learning rules (Zhou & Buonomano, 2020), providing a simple mechanism to capture the PDF.

Reply: We agree and added a section early in the Discussion in which we relate our results to existing data on interval timing from animal studies. We also discuss potential neural mechanisms that may estimate and represent the probabilistically blurred PDF as the reviewer suggests (l. 578-592).

Minor comment 5

Reviewer: I think Figure 2f-g is referenced after figure 3. Maybe Figure 3 should be 2 (since it seems to be explaining the methods used in current Figure 2)?

Reply: Thank you for pointing this out. We tried to switch Figures 2 and 3 but realized that this makes the introduction and the theoretical part on the modeling very long and resulted in a tedious read before any data are shown. We therefore opted to reference Figs. 2f-g at an earlier point in the manuscript (l. 125-127), right before the section on the modeling hypotheses. This section now transitions logically to the content displayed in Fig 3.

Minor comment 6

Reviewer: Why is the schematic of the correlation shown in Fig. 4C, if the correlation data is in Figure 5?

Reply: We see that this does not make much sense and removed the schematic from Fig. 4 and added it to Fig. 5.

Minor comment 7

Reviewer: Line 99. Provide the tau's here (as well as in the methods).

Reply: We added the tau value (it is the same for exponential and flipped exponential distributions) in the line that the reviewer names (l. 104) as well as in the methods (l. 679).

Minor comment 8

Reviewer: Line 204. "respectively" is not needed.

Reply: We removed the word.

Minor comment 8

Reviewer: Results. Provide the df's with the stats results.

Reply: We added the degrees of freedom to the appropriate analysis results (l. 114, Suppl. Table S1 & S2).

Reviewer #3

The present work aims to identify the cortical representation of temporal statistics. Human MEG and reaction time data are acquired in the context of a multimodal (auditory/visual) cued paradigm. A large arsenal of data analysis strategies specific to MEEG data, such as time and time-frequency analysis on sensor and source space, are pursued in the quest to find the cortical representation of temporal statistics. Unfortunately, the results are inconclusive and echo findings observed in different experimental contexts without manipulation of temporal expectation. The exploratory nature of the analyses is acceptable when clearly stated and

introduced, but the introduction does not justify the extensive application of various analyses to find "the cortical representation of temporal statistics."The present work will benefit from structuring and shortening the analyses with respect to the actual hypothesis.

Comment 1

Reviewer: The assertion that the probability of occurrence of a future event is represented in a cortical region is a misconception (p. 2, lines 51-55). Probability over time cannot have a cortical representation.

Reply: We are not clear that we understand this statement. The existing literature has made compelling arguments about this point. For example, there is mounting evidence that cortex can indeed represent probability over time (Tsao et al., Nat Rev Neurosci, 2022; Nobre & van Ede, Neuron, 2023), also in the case of the hazard rate (Janssen & Shadlen, Nat Neurosci 2005; Ghose & Maunsell, Nature, 2002; Bueti et al., J Neurosci, 2010), and also in the more general case of probability over a variable of interest *over time* (Beck et al. Neuron, 2006; Ma et al., Nat Neurosci, 2006). We cited some of this work which provides a theoretical foundation for our current manuscript.

Comment 2

Reviewer: Figure 1a claims neural systems count events over time. This hypothesis is unfalsifiable and appears fictional.

Reply: The reviewer opposes the assumption that neural systems count events over time.

We note that this is not a hypothesis that we test formally in our work. The general notion that the brain counts events over time is introduced (l. 65-67) to sharpen the reader's intuition about how a probability density function might be approximated by a neural system based on two basic functions: the measurement elapsed time and the counting of events over time. This is similar to building a histogram of events. The literature offers strong evidence that the human brain performs both basic operations: elapsed time estimation (e.g. Buhusi & Meck, Nat Rev Neurosci, 2005) and counting of sensory events leading to non-symbolic quantity representations (e.g. Nieder, Curr Opin Neurobiol, 2020; Nieder, MIT Press, 2019).

This *general* idea (brain counts events over time) leads to the *specific* prediction that the human brain estimates and represents the probability density function of sensory events. This is the hypothesis we test in this work. We added a section to the Discussion in which we discuss a candidate neural mechanism that may capture the PDF based on time span encoding (l. 578-592).

Comment 3

Reviewer: The central question already assumes the existence of a cortical representation of event probability (p. 2, line 59). Consequently, the logical next step is to identify the cortical representation of temporal statistics (p. 3, line 76). This is a circular logic and the premise seems questionable. This representation shaping motor responses is overly abstract.

Reply: The reviewer states that the central question of our work assumes the existence of a cortical representation of event probability and that, therefore, our aim to identify the cortical representation of temporal statistics constitutes circular logic. The reviewer further criticizes

that our assumption that the cortical representation of event probability over time shapes motor responses is overly abstract.

We appreciate that the reviewer points out that the relationship between represented probability over time and motor preparation may have been too abstract in our manuscript. Indeed it is a vast topic how the representation of information results in the formation of a motor plan which ultimately drives movement (e.g. Anderson, Neuron, 2009). In order to reduce the abstractness, we removed a panel in Fig. 1b and simplified the presumed relationship between event probability over time and motor preparation/response by focusing on our main aspect: the representation of probability over time and its relationship to reaction times (revised Fig. 1b, l. 79-86).

We respectfully disagree with the reviewer's criticism that our approach is based on circular logic. Our manuscript indeed assumes a cortical representation of event probability. First, this assumption is strongly supported by the literature (e.g. Janssen & Shadlen, Nat Neurosci, 2005; Schoffelen, et al., Science, 2005). Second, we show in behavior that humans adapt their RT to the event PDF. It is therefore logical to assume a neural representation of the event PDF and we employ a psychometric-neurometric mapping approach that is often used in the field (e.g. Janssen & Shadlen, Nat Neurosci, 2005).

Comment 4

Reviewer: In contrast, the hypothesis that a high likelihood of an event leads to faster reactions (p. 3, line 83) is trivial and empirically derived.

Reply: We respectfully disagree with both statements. A large body of work investigates how probability shapes action timing (e.g. Janssen & Shadlen, Nat Neurosci, 2005; Nobre & van Ede, Nat Rev Neurosci 2018; Cravo et al., J Neurophys, 2011; Schoddel et al., Science, 2005). The phenomenon has been studied for a long time (Response Times, Luce, 1986; Niemi & Näätänen, Psych Bull, 1981), yet the field struggles to understand the specific mechanisms that drive fast reactions based on probability estimates, a neural function underlying many aspects of cognition. As we lay out in detail in our manuscript, temporal anticipation is based on probability estimation (Pouget et al. Nat Neurosci, 2013) and on time estimation (Gallistel & Gibbon, Psych Review, 2000), two complex functions at the core of temporal cognition. Thus the relationship between probability and RT cannot be considered trivial.

Second, our hypothesis is not empirically derived. We lay out in detail the underlying rationale (see Figs. 1 and 3). This rationale is based on our previous papers on the topic that we cite in the manuscript: Grabenhorst, et al. PNAS 2021, Grabenhorst, Michalareas, et al. Nat Commun, 2019).

Comment 5

Reviewer: The statement that the human brain represents event probability over time (p. 7, line 177) is speculative and not a factual basis for the proposed modeling work.

Reply: We are puzzled at how the reviewer arrives at this conclusion. We note that we do not make a *statement* but formulate a *hypothesis* which we extensively test on a large data set. This hypothesis is supported by our previous work (Grabenhorst, et al. PNAS 2021, Grabenhorst, Michalareas, et al. Nat Commun, 2019). We lay out the rationale behind our modeling in detail (see Figs. 1 and 3).

Comment 6

Reviewer: The preset cue baseline interval (p. 8, line 207) needs adjustment to avoid post-cue data contamination.

Reply: The reviewer suggests an adjustment of the pre-Set baseline to avoid post-Cue data contamination.

We followed the reviewer's advice and performed the respective analyses again using an earlier baseline window ($t = [-0.65, -0.25]$ s) (Fig. 4 a and b). This analysis yielded very similar results to the previous analysis using the old baseline ($t = [-0.5, 0]$ s). But the reviewer's point is well-taken and with the new baseline, we avoid post-cue data contamination.

Comment 7

Reviewer: Given a sliding window of 2 sec (0.5Hz resolution), it is unclear how the peri-button press activity is smeared in time.

Reply: It is not clear to which plot the reviewer refers to. All the TFR plots are locked to the Go cue. The time point of the button press is variable with respect to the Go cue (see RT histogram in Fig. 2c). Therefore, the relative TFR phenomena will also be smeared in time with respect to the Go cue.

With respect to our time-frequency analysis settings, for each frequency, the time window is equivalent to 5 cycles and when this exceeds 500 ms (in the lower frequencies), the window is set to 500 ms. The rest of the 2 s window is zero-padded to achieve a resolution of 0.5 Hz for alignment across the different frequencies. So the actual frequency resolution is 2 Hz at the lower frequencies. This is a typical and sufficient resolution for localizing evoked activity with acceptable smearing. For illustration, we present a plot showing the TFR locked to the button press ($t = 0$, baseline: $t = [-0.65, 0.25]$ relative to Set cue, data averaged through sensors over motor cortex, grand average of 23 participants).

As you can see in the theta range, there is a clear spectral signature of the evoked potential with a fair temporal accuracy without excessive smearing with a main envelope occurring between approximately 150 ms to 320 ms.

Comment 8

Reviewer: There is an inconsistency in the units of relative power in TFR and topographies (p. 9, Fig. 4).

Reply: Thank you for pointing this out. We realized that there was a mistake in Fig. 4a (bottom). The numerical values of the relative power plotted on the y-axis were incorrect. We corrected this in the revised Fig. 4.

Comment 9

Reviewer: p.9 line 211: “Activity related to Go cue anticipation based on task demands” is a well-established beta decrease in the motor cortex prior to a button press. Task demands are minimal beyond the instruction to press a button upon the go signal.

Reply: We agree with the reviewer that a decrease in beta power prior to a motor action is a well-established finding. This neural signal is commonly thought to represent motor preparation. However, the crucial manipulation we perform in our experiment is the distribution of the sensory stimulus in time. We observe a decrease in beta power *prior* to these distributed events. The brain can only prepare the motor action (beta decrease) before a future event if it has an estimate of when this sensory event is likely to occur. Specifically, we demonstrate that the probability density of a future sensory event modulates beta power suppression, which, to our knowledge, is a new result.

Comment 10

Reviewer: The statement that alpha suppression dynamics are consistent with an anticipation signal (p. 9, lines 240-241). Alternative explanation for this rather speculative conclusion is the fact that eye blinks are consistently performed with the button press. The authors could check using the VEOG and the HEOG channels and compute the ERP. Such analysis will likely yield a consistent blink time course in VEOG and less so in the HEOG. Accordingly, alpha power is changed in preparation and under closed eyelids. As such, this has little to do with the anticipation.

Reply: The reviewer offers the hypothesis that the eye blinks may be consistently performed with the button press. In this case, the change in alpha power would not be related to anticipation. To investigate this, the reviewer suggests to investigate VEOG and HEOG channel ERPs.

We followed the reviewer's suggestion and computed the ERP of the VEOG and HEOG channels. To do so, we locked the data to the button press ($t = 0$ s is the time point of the button press) and computed the ERP first across-trials, then across participants, within each of the four experimental conditions.

In each condition, the ERP peaks around 0.5 s after the button press (Suppl. Figs. 4 & 5). The eye blinks are therefore not consistently performed together with the button press as the reviewer suggests. Importantly, the ERP analysis demonstrates that the eye blinks were also not locked in time relative to the button press but were distributed over more than 1.5 seconds after the button press. Thus the correlation between alpha power and RT is not related to the eye blinks.

However, the reviewer's suggestion to check the time course of the eye blinks relative to the Go cue is valid. We added the results of this new analysis to the manuscript (l. 308-310, Suppl. Figs. 4 & 5, l. 887-892). We also agree with the reviewer that the phrasing of our claim of an anticipation signal at this point in the manuscript was a bit strong. We softened the claim stating that the observed alpha dynamics *suggest* an anticipation signal (l. 254).

Comment 11

Reviewer: The authors need to clarify if the observed peaks reflect within-cluster separation or harmonics (p. 10, lines 269-271).

Reply: As the reviewer suggests, we investigate whether the separation into alpha and low beta bands reflects harmonics.

As in the initial manuscript, we separated the large negative correlation cluster depicted in Fig. 5a, according to alpha (7–12 Hz) and low beta (15–22 Hz) frequency bands. As can be seen in Fig. 5b and 5c, the time-frequency-channel triplets in both frequency ranges share common channels. These common channels were removed for this analysis. Within each remaining exclusive set of alpha and low beta channels, the frequency at which Spearman's rho (averaged across frequencies and across time $t = [-0.4, 0]$ s) is numerically smallest (i.e. strongest negative correlation) was computed (Suppl. Analysis).

In the alpha range, the minimum is at $f = 7.9$ Hz. The theoretical first harmonic of the alpha cluster would be at $f = 15.8$ Hz. In the low beta range the minimum is at $f = 17.0$ Hz. There is a difference of $f = 1.2$ Hz ($17.0 - 15.8$ Hz) between the theoretical first harmonic of the alpha cluster and the observed low beta cluster minimum. Thus the low beta minimum appears to not be a harmonic of the alpha minimum, supporting the separation into alpha and low beta clusters. The reviewer's point is well-taken and we added this result to the manuscript (l. 282-287, Suppl. Analysis).

Comment 12

Reviewer: The method used does not justify ROI selection (p. 10, lines 272-274). The claim appears cherry-picked and circular without identifying a „significant cluster“ considering all possible locations (https://www.fieldtriptoolbox.org/faq/how_not_to_interpret_results_from_a_cluster-based_permutation_test/).

Reply: We thank the reviewer for pointing this out. The method that the reviewer refers to is a cluster-based permutation test across all channels, all frequencies and all time points. This analysis tested the null hypothesis that the correlation between RT and spectral power across all these dimensions is not significantly different from zero. The test rejected this null hypothesis, meaning that there is significant correlation somewhere within these dimensions. This mathematical space of significance is segmented according to predefined neighboring criteria in space, time and frequency. This segmentation identified two distinct clusters, one of positive and one of negative correlation (Fig. 5a top). By examining the correlation across frequencies, within the negative cluster, we identified two prominent distinct negative peaks (Fig. 5a bottom) (see our response to comment 11 above). One is in the alpha range, the other in the lower beta range which correspond to physiologically meaningful frequency bands. The positive cluster covers the high beta frequency range. So we selected these three frequency bands in order to investigate the location of this peak correlation in source space, and in this

investigation all possible locations were considered. This peak correlation analysis was performed without any constraint on or prior selection of source space locations. We revised the manuscript accordingly (l. 357-363) and removed the words "significant cluster" where they are inappropriate. We also added a section to Supplementary Methods that explains in detail the rationale behind the peak correlation analysis ("Source-level analysis of alpha and low beta power correlation with RT.").

Comment 13

Reviewer: The statement that μ is not modulated by anticipation due to a lack of significant correlation (p. 11, lines 291-293) is incorrect. It merely indicates no statement can be made regarding the hypothesis.

Reply: We agree and deleted the claim in the manuscript (l. 303-304).

Comment 14

Reviewer: The negative correlation within the first 300 ms post-cue (p. 11, lines 295-297) might be associated with evoked activity. Concluding this effect reflects ongoing oscillation and functional implementation in temporal anticipation is premature. The spectral decomposition of evoked activity likely contributes to the observed frequency band (p. 12, lines 302-303).

Reply: The reviewer suspects that the negative correlation after the Go cue is driven by evoked activity rather than reflecting ongoing oscillation.

For the alpha band, the time-frequency representation window is around 500 ms. Given that the window is also Hanning-tapered, the effect of the ERP should start entering the window when it is centered at around $t = -100$ ms relative to the Go cue. When the sliding Hanning window is centered at $t = 100$ ms, most of the ERP activity should be contained. When the window is centered around $t = 200$ ms, the strongest ERP activity should be captured by the main lobe of the Hanning taper. The effect of the ERP will continue to be in the window at least until it is centered at approximately $t = 400$ ms.

As can be seen in Fig. 5c (bottom), up to $t = -100$ ms, there is a strong negative gradient in the correlation, reaching a maximum plateau at this time point. Up to this point, the negative correlation cannot be explained by the ERP.

After $t = -100$ ms, the negative correlation remains stable until $t = 150$ ms after the cue. Thereafter, the negative correlation is reduced to lower values. If the correlation in this range was driven by the ERP, it should remain at large negative values until much later, i.e. 400 ms. This is not seen in the data which supports the view that the negative correlation is not driven by the ERP. Furthermore, the topography of the sensors for this frequency range does not reflect sensory areas where an ERP effect would be prominent.

The low beta band case suggests a similar rationale: Here the sliding window is shorter, i.e. approximately 277 ms (5 cycles of 18 Hz). This means that the effect of the ERP should begin at around $t = -40$ ms. Given that the window is Hanning tapered, the effect of the ERP should start affecting the power estimates around $t = 0$ s. The entire ERP should be inside the window by $t = 200$. And the ERP should remain within this window until approximately $t = 400$ ms.

As can be seen in Fig. 5d (bottom), up to $t = 0$ ms, there is a sustained negative correlation. Thereafter, it drops towards zero. If the correlation in this range was driven by the ERP, it should remain at large negative values until much later, i.e. 400 ms. This is not seen in the data which

supports the view that the negative correlation is not driven by the ERP. Furthermore, the topography of the sensors for this frequency range does not reflect sensory areas where an ERP effect would be prominent.

Taken together, we lay out where in time there may be a contribution of the ERP to the spectral power estimates. However, based on the rationale outlined above the contribution to the correlation appears to be small.

Comment 15

Reviewer: The relationship between RT and alpha-beta modulations before button press (p. 12, lines 314-318) replicates earlier work and is commonly accepted. The conclusion that this relates to prediction and temporal anticipation is premature.

Reply: To the best of our knowledge, our manuscript is the first study that investigates the neural dynamics of anticipatory behavior in such a systematic way (two different probability distributions, two sensory modalities, a large data set). We are, of course, aware of the current literature on alpha and beta modulation before a motor action. That is why we investigated these neural signatures not only before a motor action, but before an anticipated sensory cue to which participants respond. We use a targeted experiment to investigate how the probability over time of a sensory Go cue modulates spectral power. Our task requires participants to predict the timing of the future Go cue. And the reaction time analysis demonstrates that participants predict based on their estimate of event probability over time. In the neural data, before the anticipated Go cue, we identified signals that predict RT to these anticipated Go cues. Critically, these signals are correlations between spectral power and *RT before the Go cue* (which is distributed in time and requires prediction as evidenced by the RT analysis). These neural signals predict participants' RT and reflect the neural dynamics that underlie the generation of RT. This is quantitative evidence that these signals are anticipation signals. We are therefore confident that our experimental design allows for identification of prediction signals and we describe this rationale throughout the manuscript (e.g. l. 433-442).

Comment 16

Reviewer: The conclusion that peri-button press changes in spectral power represent the neural representation of event PDF underlying anticipation (p. 12, lines 319-322) is too strong and unsupported by the data.

Reply: The reviewer criticizes that the conclusion that spectral power changes represent a neural representation of the event PDF is unsupported by the data.

We respectfully disagree and demonstrate (also see our reply to comment 15 above):

- i) The dynamics of alpha and beta oscillations *prior to the Go cue* predict the timing of the button press *after the Go cue*. i.e. RT. Since RT dynamics follow the event PDF (Fig. 2d and e) this is evidence for a neural representation of the event PDF, i.e. a prediction signal prior to the Go cue.
- ii) We show that neural activity in alpha and beta frequency bands is directly correlated with with the fit event PDF-based model (Suppl. Fig. 8).

Comment 17

Reviewer: The frequency resolution suggested by DICS method results (p. 13, line 340) appears inconsistent. A window of 100 ms results in Δf of 10 Hz. How were 7,8,9 Hz etc. estimated?

Reply: All sensor-level spectral complex coefficients for all time points (20 ms resolution) and trials were projected through the DICS spatial filters into source space. There the power was computed from the amplitude of each projected complex coefficient. These power values were then averaged within frequency bands-of-interest and time windows of 100 ms, centered at $t = [-0.35, -0.25, -0.15, -0.05, 0.05, 0.15, 0.25, 0.35]$ s relative to the Go cue. This was done for each trial. Then the average power value per window and frequency band was Spearman-correlated with RT across trials. So the actual frequency resolution is the same as in the sensor-level data. As already mentioned, zero-padding was used to derive the same resolution of the frequency axis for all frequencies.

We realized that this aspect of our analysis requires clarification in the manuscript and added a section to the Methods (l. 865-871).

Comment 18

Reviewer: The section discussing significant clusters is redundant and circular (p. 13). The authors already rejected the H0 on sensor level and yet using the same data the hypothesis is used again to evaluate effects on source level. Acknowledging this fact and avoiding overstatement is recommended (i.e. "After 0.15 s post cue, there were no significant clusters.")

Reply: We revised the manuscript and added a section that clarifies that we do not again reject the H0 on source level (l. 357-363). Throughout the manuscript, we now differentiate between the rejection of H0 at the sensor level and our peak correlation analysis which is aimed to identify the cortical location of the correlation in specific frequency bands. We further lay out the rationale behind this in Supplementary Methods (see our reply to comment 12 above). We also removed the sentence that the reviewer pointed out (l. 369).

Comment 19

Reviewer: Similar sensor and source maps have been reported in other contexts, such as the HCP WM task (p. 13-15). For example <https://pubmed.ncbi.nlm.nih.gov/29803957/>. The right lateralization of the effects appears not to be specific to the "representation of event PDF" or the "supra-modal functional role of SPL in temporal anticipation."

In fact, it appears that the authors could perform the same analyses on publicly available data and arrive at a similar conclusion, despite the fact that event anticipation, temporal predictions, and sensory modality were not manipulated.

Reply: If we understand correctly, the reviewer says that different cognitive functions are associated with similar sensor and source maps. We agree. It is well-established that many cortical areas subserve different cognitive functions. But it is also known that different cognitive functions require different computations. Here we investigate the computations underlying temporal anticipation in behavior and its neural underpinnings using a targeted design (see our reply to comment 15).

However, we struggle with the generalization in reviewer's assumption that we "could perform the same analyses on publicly available data and arrive at a similar conclusion, despite the fact that event anticipation, temporal predictions, and sensory modality were not manipulated."

We note that, based on *technical grounds*, one cannot perform "the same analyses on publicly available data" (as the reviewer proposes): e.g. if there is no regressor (RT to a sensory cue whose timing follows an event PDF or the event PDF itself) in such experiments, the analysis is simply not possible, thus the targeted experiment.

We also note that, based on *logical grounds*, one cannot "arrive at a similar conclusion" (as the reviewer proposes): e.g. a working memory task performed in the visual modality (the paper the reviewer cites) is not informative about temporal prediction in audition etc.

Comment 20

Reviewer: The analyses on p. 18 are also rather circular. The identification of temporal clusters and their use as TOI is not justified by the methodology.

Reply: We do not see how our analysis steps are circular and clarify the pipeline in the following:

- We first perform a sensor-level analysis where a cluster-based permutation test identifies temporal intervals in which the magnetic field from brain sources is correlated with RT.
- Having identified these intervals of significant correlation, we project the measurements from all sensors, and not just from the sensors of the clusters, through linear spatial filters to source space. There we compute again the correlation for each brain location.
- Then we run a permutation test in order to identify the areas with significant correlation and mask areas where the correlations could be due to chance.
- This sequence of analyses is not circular: No spatial information from sensor space is used in source space. Only selection of a temporal interval is made. In source space, only the spatial distribution of correlations is examined and nothing regarding the temporal structure.

We firmly believe that this analysis is valid.

ORIGINAL REVIEWERS' COMMENTS

Reviewer #1 (Remarks to the Author):

Review of Grabenhoerst Poeppel

202: "We focused the analysis on the time span preceding the Go cue (Fig. 2a) since this is when a representation of the event PDF should be reflected in neural activity."

Some amplification of the thinking is required here. Why should it occur before the activity triggered by the Go cue, given that it is the velocity of the triggered activity that is altered by knowledge of the pdf. I assume the thought is that the earlier activity is setting up the neural circuitry that will be activated by the Go signal to perform most precisely at an upcoming time.

265: "the correlation between pre-Go-cue spectral power and RT can be interpreted as a neural representation of the event PDF" Would it not be better to say that they are indicative of or a reflection of a neural representation? Does anyone believe that there is a neural process

that reads the information encoded in the fluctuations in magnetic field spectral power? I assume that these temporally and spatially localized fluctuations are indicative of relevant neural activity at those spatio-temporal locations.

273 “In each of these two negative clusters, only the significant sensors within the corresponding frequency band were selected.” I don’t understand this. Are the sensors themselves selectively tuned to specific frequencies? That’s what it seems to suggest. Or do you mean to say that the signals that they entered into their next analysis came only from those sensors located in the regions where there was a significant dip in these power bands in anticipation of the Go signals?

412: “The correlation analysis between spectral power and RT reveals that, before anticipated auditory and visual sensory events, alpha, low beta and high beta spectral power dynamics represent an approximation of the event PDF.” Make it clearer what the basis of this assertion is. The behaviorally defined pdf is specified with regard to its location in time following (not preceding) the Go signal and by its slash-like shape (Figure 2d&e). The power fluctuations are not in the same temporal location nor do they have the same shape, so one needs to explain why they may be said to approximate the behaviorally estimated pdf.

577: It is argued that the computational processes underlying such time estimation are directly linked to neural population dynamics¹⁸” Not sure what is being asserted here. Suppose for the sake of argument one imagined that the computations were carried out intracellularly by molecular machinery and that the spiking activity that (I assume) is the dominant source of MEG spectral power, transmits the results of those computations to other neurons. Would that constitute a direct linkage? What would constitute indirect linkage?

588: “One obvious question is how the observed lateralization to right parietal areas relates to the generation of a left lateralized motor command, as required by our task.” The lengthy musings that follow the posing of this question and that reappear in the Discussion serve only to emphasize how conjectural and vague are the possible inferences about areal functions that one might draw from these findings. There is nothing to be ashamed of about this because even the current story about the function of, for example, even V1 is incomplete and vague at many points, and for that region we have vastly more and more direct data on neural activity under controlled conditions. However, the tenuous and vagueness of these musings does call into question the value of including them in the publication of the findings.

The writing is inelegant. A technically sophisticated copy editor could make it better. Among the simplest edits would be to strike out the emphatic adverbs and adjectives: “Remarkably” “convincing” etc. Beyond that, simplification of sentence structure would improve the exposition.

Here for illustration is one of many awkward run-on sentences: 438 “This analysis further demonstrates that neural dynamics in alpha and beta frequency bands represent the event probability distribution over time prior to anticipated sensory events in higher-order cortex and in sensorimotor cortex.”

I recommend radically reducing the speculative remarks about functionality and dialing back—or clarifying—the claim that the spatio-temporally localized fluctuations at certain frequencies of the spectral power approximate the behaviorally estimated pdf. To my mind, what these fluctuations indicate is simply where to look for more direct data on how the pdf is derived and how it controls the reaction time. The complexity of the correlational analyses and the multitude of variables that come into play make it hard to extract a clear take home message.

C. R. Gallistel

Reviewer #2 (Remarks to the Author):

This paper uses an implicit timing task (i.e., one in which the task itself—pressing a button in response to a stimulus—does not rely on explicit temporal information). The standard view is that during implicit timing tasks subjects create a temporal expectation of when the stimulus will arrive based on the hazard function (the probability an event is about occur given that it has not yet occurred). Here the authors provide compelling evidence that temporal expectation is not based on the hazard rate, but the PDF (probability density function, i.e., the distribution of the experienced intervals). The authors show that reaction times (RTs) are not quantitatively (or in some cases qualitatively) captured by the hazard rate, but well captured by the PDF. MEG studies demonstrated a supramodal inverse correlation between RT and alpha power in the R superior parietal cortex, and a correlation between beta power and RT in the L sensorimotor cortex. The paper significantly advances, and alters, our current understanding of this form of implicit timing, and provides significant insights into the potential brain areas contributing to timing and the encoding of the PDF.

My major comments are:

1. It was not clear why the PDF model used the probabilistic blurring rather than temporal blurring as in the hazard rate model. Is the probabilistic blurring supposed to capture the perceptual learning of the intervals (which generally does not happen quickly)? As it currently stands, the reader can fairly ask how much of the difference between the models stems from the use of temporal blurring in the hazard mode versus probabilistic blurring in the PDF model? It seems necessary to perform an apples-to-apples comparison of both models using the same blurring function—either the hazard with the probabilistic blurring or the PDF with the temporal blurring. As I understand it, using the same blurring function in both models should not change the results that much?

2. The abstract states “These results show that supra-modal representations of probability density across cortex underlie the anticipation of future events”. To arrive at the conclusion that the observed MEG signals represent the PDF wouldn’t it be necessary to directly contrast the alpha and beta dynamics during the exponential and flipped distributions? Relatedly, can might the alpha and beta dynamics not be encoding the PDF but elapsed time itself—that is, can elapsed time from the Set cue be decoded from changes in alpha/beta power?

Additional comments and suggestions:

While the correlation values between alpha/beta power and RT are significant they seem fairly weak. Is this normal in these types of MEG analyses? Is it possible to perform correlations between the phase of alpha and RTs (e.g., Lakatos, ..., Schroeder, 2008).

Figure 2 provides a nice analyses of the RT results with the exponential and inverse exponential PDFs. However, RT is often inversely correlated with accuracy. Was there a significantly number of false (early) responses? If so where there any differences in accuracy (as measured by false response) between modality and distribution, and on the timing of these false responses between the distributions?

In the Discussion it would be helpful to better place the current results in the context of existing models and the large amount of data on timing from animal studies. Particularly if it turns out that one can decode time from the alpha/beta dynamics. A key point of the paper is the PDF is simpler to calculate and thus more biologically plausible. Indeed, it has been pointed out that the orthogonality of neural sequences are well suited to read out time using associative learning rules (Zhou & Buonomano, 2020), providing a simple mechanism to capture the PDF.

I think Figure 2f-g is referenced after figure 3. Maybe Figure 3 should be 2 (since it seems to be explaining the methods used in current Figure 2)?

Why is the schematic of the correlation shown in Fig. 4C, if the correlation data is in Figure 5?

Line 99. Provide the tau's here (as well as in the methods).

Line 204. "respectively" is not needed.

Results. Provide the df's with the stats results.

Reviewer #3 (Remarks to the Author):

The present work aims to identify the cortical representation of temporal statistics. Human MEG and reaction time data are acquired in the context of a multimodal (auditory/visual) cued paradigm. A large arsenal of data analysis strategies specific to MEEG data, such as time and time-frequency analysis on sensor and source space, are pursued in the quest to find the cortical representation of temporal statistics. Unfortunately, the results are inconclusive and echo findings observed in different experimental contexts without manipulation of temporal expectation. The exploratory nature of the analyses is acceptable when clearly stated and introduced, but the introduction does not justify the extensive application of various analyses to find "the cortical representation of temporal statistics." The present work will benefit from structuring and shortening the analyses with respect to the actual hypothesis.

The assertion that the probability of occurrence of a future event is represented in a cortical region is a misconception (p. 2, lines 51-55). Probability over time cannot have a cortical representation.

Figure 1a claims neural systems count events over time. This hypothesis is unfalsifiable and appears fictional.

The central question already assumes the existence of a cortical representation of event probability (p. 2, line 59). Consequently, the logical next step is to identify the cortical representation of temporal statistics (p. 3, line 76). This is a circular logic and the premise seems questionable. This representation shaping motor responses is overly abstract.

In contrast, the hypothesis that a high likelihood of an event leads to faster reactions (p. 3, line 83) is trivial and empirically derived.

The statement that the human brain represents event probability over time (p. 7, line 177) is speculative and not a factual basis for the proposed modeling work.

The preset cue baseline interval (p. 8, line 207) needs adjustment to avoid post-cue data contamination. Given a sliding window of 2 sec (0.5Hz resolution), it is unclear how the peributton press activity is smeared in time.

There is an inconsistency in the units of relative power in TFR and topographies (p. 9, Fig. 4). p.9 line 211: "Activity related to Go cue anticipation based on task demands" is a well-established beta decrease in the motor cortex prior to a button press. Task demands are minimal beyond the instruction to press a button upon the go signal.

The statement that alpha suppression dynamics are consistent with an anticipation signal (p. 9, lines 240-241). Alternative explanation for this rather speculative conclusion is the fact that eye blinks are consistently performed with the button press. The authors could check using the VEOG and the HEOG channels and compute the ERP. Such analysis will likely yield a consistent blink time course in VEOG and less so in the HEOG. Accordingly, alpha power is changed in preparation and under closed eyelids. As such, this has little to do with the anticipation.

The authors need to clarify if the observed peaks reflect within-cluster separation or harmonics (p. 10, lines 269-271).

The method used does not justify ROI selection (p. 10, lines 272-274). The claim appears cherry-picked and circular without identifying a „significant cluster“ considering all possible locations

(https://www.fieldtriptoolbox.org/faq/how_not_to_interpret_results_from_a_cluster-based_permutation_test/).

The statement that μ is not modulated by anticipation due to a lack of significant correlation (p. 11, lines 291-293) is incorrect. It merely indicates no statement can be made regarding the hypothesis.

The negative correlation within the first 300 ms post-cue (p. 11, lines 295-297) might be associated with evoked activity. Concluding this effect reflects ongoing oscillation and functional implementation in temporal anticipation is premature.

The spectral decomposition of evoked activity likely contributes to the observed frequency band (p. 12, lines 302-303).

The relationship between RT and alpha-beta modulations before button press (p. 12, lines 314-318) replicates earlier work and is commonly accepted. The conclusion that this relates to prediction and temporal anticipation is premature.

The conclusion that peri-button press changes in spectral power represent the neural representation of event PDF underlying anticipation (p. 12, lines 319-322) is too strong and unsupported by the data.

The frequency resolution suggested by DICS method results (p. 13, line 340) appears inconsistent. A window of 100 ms results in Δf of 10 Hz. How were 7,8,9 Hz etc. estimated?

The section discussing significant clusters is redundant and circular (p. 13). The authors already rejected the H_0 on sensor level and yet using the same data the hypothesis is used again to evaluate effects on source level. Acknowledging this fact and avoiding overstatement is recommended (i.e. "After 0.15 s post cue, there were no significant clusters.")

Similar sensor and source maps have been reported in other contexts, such as the HCP WM task (p. 13-15). For example <https://pubmed.ncbi.nlm.nih.gov/29803957/>. The right lateralization of the effects appears not to be specific to the "representation of event PDF" or the "supra-modal functional role of SPL in temporal anticipation." In fact, it appears that the authors could perform the same analyses on publicly available data and arrive at a similar conclusion, despite the fact that event anticipation, temporal predictions, and sensory modality were not manipulated.

The analyses on p. 18 are also rather circular. The identification of temporal clusters and their use as TOI is not justified by the methodology.

We again thank the reviewers for their constructive comments and reply in in detail.

Reviewer #1

Comment 1

Reviewer: Line 219: "...that neural activity before an anticipated cue may prepare the neural circuitry that will be activated by the cue...." I suggest that 'potentiate' is better (more specific) than 'prepare'

Reply: We agree and changed the sentence accordingly.

Comment 2

Reviewer: Line 441: 'before to' ungrammatical

Reply: We corrected the sentence.

Comment 3

Reviewer: p. 584-586: This sentence treats temporal probability as an alternative to the well established scalar variability in the estimation of elapsed intervals. While it may be possible to elaborate a model in which this is the case, they do not do so here, and, unless my intuitions are leading me seriously astray, such a model would have to make some pretty strong assumptions, which would seem to me highly implausible. The subject's measure of the time elapsed presumably provides the support vector for the subject's probability density function in these experiments, where they imposed two different (mirror imaged exponential) probability density functions on the elapsed time at which the Go signal would occur. My own conception, which I had assumed was the authors' conception, is that the subjects impose corresponding response-potential functions on their measure of the interval elapsed after the Set signal. Thus, when the Go-probability steps up to its maximum at 0.4 s after the Set signal, the response to the Go signal is maximally potentiated and therefore occurs with the minimum latency when the subject's measure of time elapsed reaches 0.4s. When the Go-probability steps up to a minimal probability at 0.4s and then accelerates to a maximum at 1.4s, the subject's response-potential function has the same profile, but again, this profile is supported by the subject's measure of the elapsed interval.

I haven't thought this out carefully but if my intuitions are right, then the subject's function will be the convolution of the experimenter-imposed exponential or flipped exponential and the subject's Gaussian distribution on elapsed time, which is parameterized by its mean, which equals the actual time elapsed, and a constant(!) cov, hence an uncertainty (variability) proportionate to elapsed time (scalar variability). Intuitively, the horizontal variability in the support vector must spread out the probability density function. For a model to claim that the only factor is the pdf on time, it would have to assume that variability in the support vector did not increase over time, which seems to me implausible. More importantly, there is a lot of experimental evidence against [it.]

Reply: The reviewer has absolutely valid concerns about making strong claims against the validity of the scalar property of elapsed time, which has been well established by a large body of experimental work. Our contribution here regarding the effect of the probability

distribution on the uncertainty in time estimation should be seen in the context and limitations of the specific experimental design.

The time intervals investigated here are bounded in the range [0.4 1.4] seconds. These intervals of elapsed time are considered short. In the seminal opinion paper by Gibbon, Malapani, Dale and Gallistel 1997 in Phil. Trans. of Royal Society B, the range between 0.1 and 1.5 is considered as a range with a distinct pattern of the Coefficient of Variation (CV). In this range, the CV has a “dip” with its smallest value. This means that the standard deviation in the estimation of elapsed time is increasing relatively slowly with elapsed time in this interval. Our results offer some evidence that, in this specific range, when there is a very high event probability density in the later part of the interval, it can potentially override, to a certain extent, the slowly increasing variance.

First of all, we have no evidence yet that this effect can occur at longer elapsed time intervals. Especially as for longer intervals the CV is increasing significantly. Second, the evidence for the better fitting of a model comes mainly from the later end of the distribution with the highest event probability (compare Figs. 2d (right) and 2e (right) with the corresponding plots in Suppl. Fig. 3b). But this evidence cannot speak for a complete overriding of the scalar property by probability density, especially for time intervals outside of the range of $t = [0.4, 1.4]$ s.

More targeted experiments are required to understand to what extent and in what ranges, different probability densities can distort the scalar property of time estimation. It is very likely that such effect would be best described by a model incorporating both of these sources of variance. However, we strongly believe that for the purpose of the current study, in this range of short intervals, it is an important finding that the data can be better described by a model where the uncertainty in estimation of elapsed time is modulated by high probability density.

Figure 3

Coefficient of variation in timing tasks as a function of the magnitude of the interval being timed (both axes on log scales). In the ranges demarcated by vertical lines, coefficients of variation appear to initially decrease (less than 0.1 s to greater than 0.1 s), remain roughly constant for given tasks between 0.1 s and about 1.5 s, increase perhaps again between 1.5 s and 500 s, then increase again for times in the hours range at about 500 s. Large points denote studies spanning these ranges. Open points denote human studies, filled points animal studies.

Reproduced from Gibbon, Malapani, Dale and Gallistel 1997 for convenience.

Reviewer #2

The authors have done an excellent job in addressing my concerns. In particular it is very helpful to know that it was not possible to decode elapsed time from alpha/beta, and that there was no information in the phase. It is also now much clearer that the correlations were performed across the four relevant conditions, making the reported findings of correlates of the PDF in the IPL, pMTG important and compelling. Overall the authors have gone above and beyond in addressing the concerns of all reviewers concerns.

Reply: Thank you for offering your advice on how to improve our manuscript and for recognizing our efforts.

Reviewer #3

I was wrong. I have been taught a lesson on various levels.

I appreciate the clarity, depth, and structure of the response letter. I follow the author's arguments. I have realized that without providing a context, my critique achieved nothing but appeared cynical at best. This wasn't my intention, and I apologize to the authors.

I was coming from a standpoint of a general problem in the field assigning various sorts of psychological functions to often similar, sometimes the same voxels, brain rhythms, and time points. Doing so, I put words in the author's narrative, which is naively foolish. Pursuing research within the accepted premises and current standards in the field, as the present work clearly does, is the scientific method and cannot be criticized without context. Which I have not provided. The review process is, however, neither the forum nor the format to pursue this debate further.

I appreciate the additional work RE EOG and blinks.

My point was that blinks might be consistent with the button press, not zero-phase locked to the button press. In my reading, the supplemental figure demonstrates this. Their execution might be associated with similar preparatory alpha/beta as is the case for button presses. If the latter is correlated with blinks as it appears to be, one would also expect the correlation between alpha and RT. One could median split the trials concerning the EOG blink amplitude (or some sort of percentiles, even trials where no blink was executed after the button press maybe). In this case, the relationship reported in Figure 5c shouldn't change. If it does, however, I feel that the conclusion that the correlation is not driven by (a preparation for) eye blink activity entailed in the motor act button press could be revisited. I do not suggest the authors should do this in the present work. Just the context for the point I had raised.

Additional revision rounds appear, in my view, not necessary. I recommend publication. Thank you.

Reply: We thank the reviewer for acknowledging our efforts. The reviewer raises again the valid point that the eye blinks might be systematically related to the time point of the button press. (We assume that this is what the reviewer means by saying "consistent with the button press".)

Supplementary Figures 4 & 5 show the eye-blink muscle ERPs as measured with electrode channels EOGV and EOGH. The muscle ERPs cover a time span of approx. 1.5 s which is much longer than a typical eye blink muscle ERP. We therefore argue that the peaks of these plots do not indicate that the eye-blinks are locked to the button press but that they reflect the distribution of the eye-blink muscle ERPs over time. In order to investigate the distribution of the eye blinks relative to the button press further, we performed two more analyses.

First, at the single-trial level, we computed the time points at which the EOGH channel's signal is maximal over time span $t = [0, 1.75]$ s relative to the button press. The histogram of these time points demonstrates that the single-trial eye blinks were indeed distributed over time (Supplementary Fig. 6).

Second, we analysed eye-tracking data. At the single-trial level, we computed the time points when the eye is closed (i.e. "pupil diameter is minimal") over $t = [0, 1.75]$ s. This is a different measure of when the eye blink occurred relative to the button press (Supplementary Fig. 7). This analysis confirms the results of the first analysis.

Taken together, the eye blinks are distributed in time relative to the button press. Thus the correlation between spectral power before the Go cue and RT is not related to the eye blinks.